# Hierarchical TAF1-dependent co-translational assembly of the basal transcription factor TFIID

Andrea Bernardini [1,2,3,4], Pooja Mukherjee[1,2,3,4,6], Elisabeth Scheer[1,2,3,4], Ivanka Kamenova[1,2,3,4,7], Simona Antonova[5,8], Paulina Karen Mendoza Sanchez [5], Gizem Yayli[1,2,3,4], Bastien Morlet[1,2,3,4], H.T. Marc Timmers [5] & László Tora [1,2,3,4] ✉

Large heteromeric multiprotein complexes play pivotal roles at every step of gene expression in eukaryotic cells. Among them, the 20-subunit basal transcription factor TFIID nucleates the RNA polymerase II preinitiation complex at gene promoters. Here, by combining systematic RNA-immunoprecipitation (RIP) experiments, single-molecule imaging, proteomics and structure–function analyses, we show that human TFIID biogenesis occurs co-translationally. We discovered that all protein heterodimerization steps happen during protein synthesis. We identify TAF1—the largest protein in the complex—as a critical factor for TFIID assembly. TAF1 acts as a flexible scaffold that drives the co-translational recruitment of TFIID submodules preassembled in the cytoplasm. Altogether, our data suggest a multistep hierarchical model for TFIID biogenesis that culminates with the co-translational assembly of the complex onto the nascent TAF1 polypeptide. We envision that this assembly strategy could be shared with other large heteromeric protein complexes.

Large heteromeric protein complexes are implicated in all aspects of gene expression, and uncovering their assembly mechanism is particularly challenging. Several different subunits, synthesized by separate mRNA molecules, must productively interact with their direct partners in the crowded cellular environment and sequentially build larger assemblies while minimizing off-pathway interactions and aggregation.

Co-translational assembly (co-TA) can facilitate the formation of protein complexes, whereby the newly synthesized nascent protein chain establishes the interaction with the partner before it is released from the ribosome[1,2]. Co-TA can be sequential (also termed directional), which is when a fully translated protein interacts with the partner nascent chain, or simultaneous (also termed symmetrical), which is when both interactors are nascent chains. Coupling specific subunit–subunit assembly with translation would reduce the exposure of aggregation-prone domains, facilitate the formation of intricate protein–protein interfaces and allow a sequential order for the assembly of different subunits[1,3]. Co-TA participates in the heterodimerization of several yeast proteins[4–7], including the assembly of subunits of the nuclear pore complex[8,9].

Many of the molecular machines involved in transcription initiation are large heteromeric protein complexes. Among them, the ~1.3-MDa basal transcription factor TFIID makes contacts with core promoter DNA elements, promotes TATA-binding protein (TBP) loading on core promoters and works as a scaffold for the formation of RNA polymerase II preinitiation complex (PIC) on all protein-coding genes[10–12].

[1]Institut de Génétique et de Biologie Moléculaire et Cellulaire, Illkirch, France. [2]Centre National de la Recherche Scientifique, Illkirch, France. [3]Institut National de la Santé et de la Recherche Médicale, Illkirch, France. [4]Université de Strasbourg, Illkirch, France. [5]German Cancer Consortium (DKTK) partner site Freiburg, German Cancer Research Center (DKFZ) and Department of Urology, Medical Center-University of Freiburg, Freiburg, Germany. [6]Present address: Innovative Genomics Institute, University of California, Berkeley, CA, USA. [7]Present address: Nature Protocols, London, UK. [8]Present address: The Netherlands Cancer Institute, Amsterdam, the Netherlands. ✉e-mail: laszlo@igbmc.fr

In metazoans, TFIID comprises TBP and 13 TBP-associated factors (TAFs)[13], and it can be subdivided into three structural lobes (Fig. 1a)[14,15]. Single-particle cryo-electron microscopy (cryo-EM) models of yeast and human TFIID have shed light on the position and atomic interactions among its subunits[10,16–18].

Nine TAFs contain a histone-fold domain (HFD) that dictates five defined dimerization interfaces within the complex, namely TAF4–TAF12, TAF6–TAF9, TAF3–TAF10, TAF8–TAF10 and TAF11–TAF13. A set of five TAFs (TAF4, TAF5, TAF6, TAF9 and TAF12)—named core-TFIID—is present in two copies, constituting a pseudo-symmetrical unit within the complex that occupies both A and B lobes. These two lobes differ in the dimerization partner of TAF10: TAF3 in lobe A, and TAF8 in lobe B. Moreover, lobe A is characterized by the additional TAF11–TAF13 HFD pair (Fig. 1a). Lobe C comprises the structured domains of the TAF1–TAF7 dimer, TAF2 and the central HEAT domains of the two copies of TAF6. The B and C lobes are connected through the carboxy-terminal portion of TAF8, which directly interacts with TAF2 (ref. 19), whereas the A lobe remains flexibly connected through the TAF6 linker region.

How and where TFIID assembles in cells is a longstanding question. Classically, the holocomplex is isolated from nuclear extracts, while attempts to isolate endogenous assemblies in the cytoplasm led to the identification of preformed TAF2/TAF8/TAF10 and TAF11/TAF13 modules[20,21]. Another hint on the formation of cytoplasmic TFIID submodules came from the isolation of a stable TAF5/TAF6/TAF9 subcomplex[22]. These observations led to a model whereby different TFIID modules would be formed in the cytoplasm and holocomplex formation would take place in the nucleus[23].

We have demonstrated co-TA events between three pairs of TFIID subunits in the cytoplasm, either sequential (TAF10–TAF8, TBP–TAF1) or simultaneous (TAF6–TAF9)[24]. With the aim of searching for additional co-TA events in TFIID, we carried out a broad combination of complementary approaches, and we identified a series of pairwise co-TA events that shape the early steps of TFIID assembly. Unexpectedly, we uncovered a new role for the TAF1 nascent protein as a co-translational 'landing platform' for preassembled TFIID submodules in the cytoplasm.

## Results

### Co-translational interactions in TFIID identify TAF1 as a central hub

When reanalyzing our previously published TAF10 RNA immunoprecipitation (RIP)-microarray data[24], *TAF1* mRNA scored as a positive hit (Extended Data Fig. 1a). TAF1 is devoid of HFDs and it is not known to directly interact with TAF10 within TFIID, raising the possibility that higher-order co-translational interactions might take place. This observation prompted us to perform TAF10 RIP–qPCR on HeLa cells polysome extracts (Fig. 1b). Indeed, we detected a strong enrichment of *TAF1* mRNA in TAF10 RIP assays, along with the expected mRNAs of *TAF10* itself and its HFD partner *TAF8*. *TAF1* mRNA enrichment was reproducible and puromycin-sensitive, suggesting a co-translational association of TAF10 with the nascent TAF1 polypeptide. To rule out biases from the cellular system or the antibody, we performed the same experiment on E14 mouse embryonic stem cells (mESCs) using a different monoclonal antibody than the one used in the human system. We found that the anti-TAF10 RIP enriched *Taf1* mRNA in mouse cells as well (Fig. 1c), suggesting that the phenomenon is conserved.

To systematically assess all co-translational assembly events within the TFIID complex, we used a series of inducible HeLa cell lines engineered to express each TFIID subunit as a fusion protein with an amino-terminal GFP tag[25]. These GFP-tagged TAFs have been shown to be incorporated into TFIID purified from nuclear extracts[26]. We performed GFP-RIP assays on polysome extracts for each individual TFIID subunit and used RT–qPCR to systematically test for enrichment of mRNAs encoding all the TFIID subunits (Fig. 1d). The results of this systematic RIP–qPCR screening confirmed the previously published TFIID co-TA

subunit pairs (TAF10–TAF8, TAF6–TAF9 and TBP–TAF1), validating the general reliability of the system (Fig. 1e and Extended Data Fig. 1b). Strikingly, our systematic assay revealed that *TAF1* mRNA was enriched in RIP experiments of several distinct TFIID subunits (Fig. 1e). TAF10 RIP also scored positive for *TAF1*, confirming the observations from endogenous TAF10 RIP assays (Fig. 1b–c). Apart from TAF10, RIP assays of TAF2, TAF4, TAF5, TAF8, TAF12 and TBP retrieved *TAF1* mRNA (Fig. 1e).

In addition, novel subunit pairs undergoing co-TA were detected, including well-established HFD partners: TAF10 interacts co-translationally with nascent TAF3; TAF12 with nascent TAF4; and TAF11 and TAF13 are reciprocally enriched, hinting at symmetrical co-TA (Supplementary Table 1). Our systematic RIP assay also revealed co-TA among direct partner subunits that do not interact through an HFD. For instance, TAF2 and TAF8, which are known to interact directly in TFIID (Fig. 1a and Extended Data Fig. 1c), reciprocally enriched the partner's mRNA, suggesting simultaneous co-TA. TAF5 enriched *TAF6* mRNA, one of its direct interactors within core-TFIID: TAF6 contributes with a β-strand to the last blade of the TAF5 WD40 β-propeller domain (Extended Data Fig. 1d). Finally, TAF1 enriched the mRNA of *TAF7*, its direct partner within TFIID (Fig. 1e).

We noted that, for three of the GFP-fusion-protein-expressing cell lines (TAF6, TAF13 and TBP), we could not retrieve the bait mRNA in our RIP assays, and that the anti-GFP-TAF7 RIP failed to retrieve *TAF1*. TBP–TAF1 co-TA has already been shown with endogenous TBP RIP assays in our previous report[24]. To complete our systematic screening, we performed RIP assays with antibodies recognizing endogenous TAF6 and TAF7 and observed a robust puromycin-sensitive enrichment of *TAF1* mRNA in both TAF6 and TAF7 RIP experiments (Fig. 1f,g and Extended Data Fig. 1e). Overall, these observations expand the repertoire of TFIID subunits that follow the co-TA pathway with their partners, and importantly identify the nascent TAF1 protein as a potential hub for the recruitment and assembly of many TFIID subunits.

### TAFs are localized in the proximity of *TAF1* mRNA in the cytoplasm

Our observations made on the basis of RIP assays suggest that, during *TAF1* mRNA translation, several TFIID subunits physically associate with TAF1 nascent polypeptide. To physically localize and quantify these events in an endogenous cellular context, we combined immunofluorescence (IF) against several TFIID subunits with single-molecule RNA fluorescence in situ hybridization (smFISH) using HeLa cells.

First, we used this strategy to detect TAF1 nascent protein and estimate the fraction of actively translated *TAF1* mRNAs. To this end, we used an IF-validated TAF1 antibody recognizing an N-terminal antigen and combined it with *TAF1* mRNA smFISH (Fig. 2a). We used *CTNNB1* as a negative control mRNA in smFISH. The average number of cytoplasmic mRNAs per cell for *TAF1* was about 16, and it was about 120 for *CTNNB1* (Fig. 2b). Next, we combined TAF1 IF with *TAF1* or *CTNNB1* smFISH (Fig. 2c) and quantified the number of *TAF1* mRNA molecules co-localizing with TAF1 protein spots. About ~55% of *TAF1* cytoplasmic mRNAs co-localized with TAF1 IF spots (Fig. 2d). This fraction decreased by more than tenfold upon puromycin treatment, proving a dependence on mRNA, ribosome and nascent chain integrity. The very low fraction (~1%) of co-localization with *CTNNB1* mRNA was puromycin-insensitive and represents random co-localization. We also validated the specificity of the *TAF1* smFISH signal by short interfering RNA (siRNA)-mediated knockdown of *TAF1* (Extended Data Fig. 2). Roughly half of *TAF1* mRNAs were detected as being actively translated in HeLa cells. However, nascent protein detection on poorly translated mRNAs might still be missed.

Next, we applied the same strategy to TFIID subunits to assess their spatial proximity to *TAF1* mRNA (Fig. 3a). First, we assessed the combination with TBP (lobe A component) (Fig. 3b), as its co-translational association with TAF1 has already been dissected[24]. We found that ~6% of cytoplasmic *TAF1* mRNAs co-localized with TBP, whereas less than

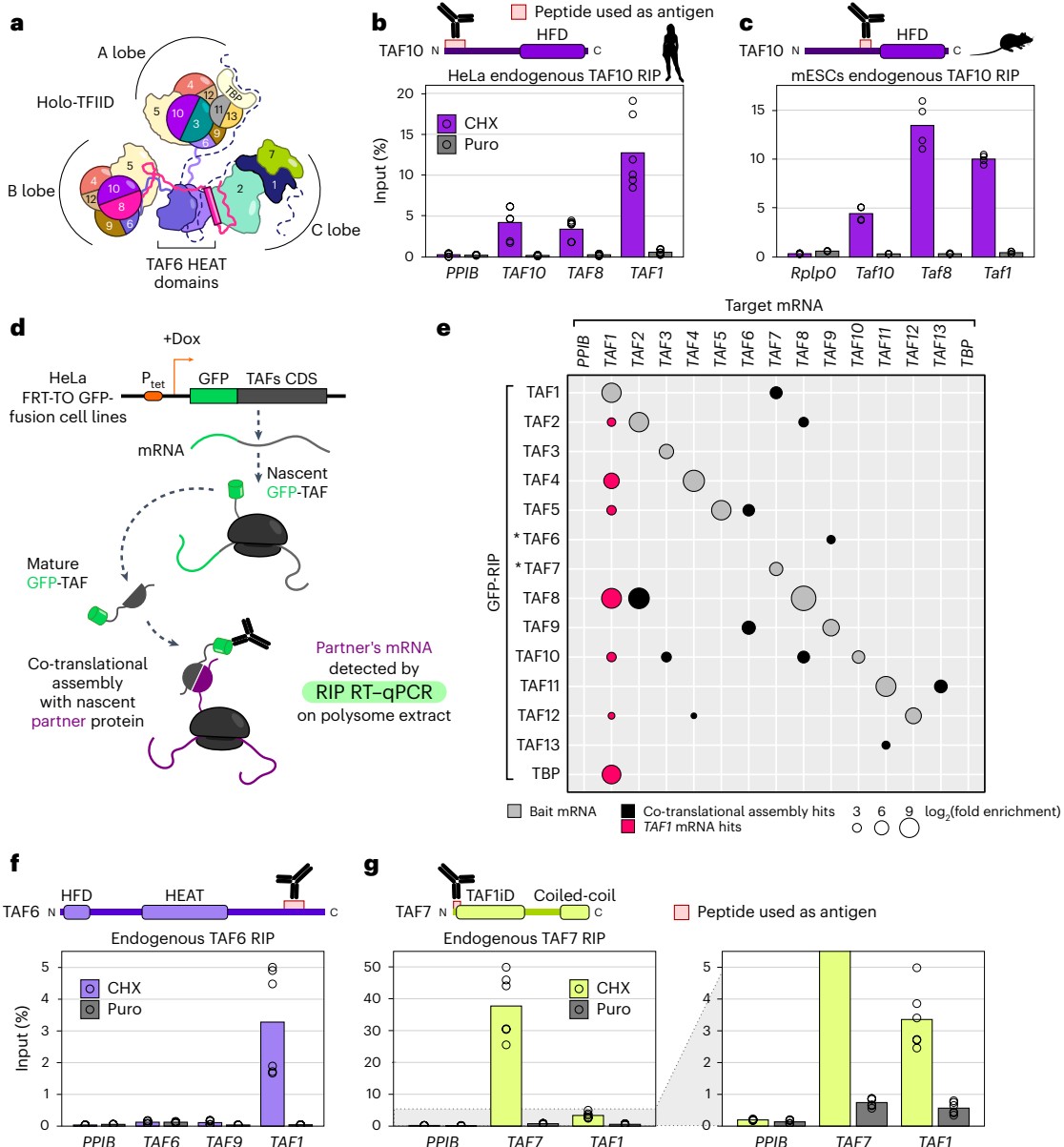

**Fig. 1 | A systematic assay expands the network of co-translational interactions in TFIID and identifies nascent TAF1 polypeptide as a central hub in the assembly process. a**, Schematic structure of TFIID. Half-circle subunits represent HFD partners. **b**, RNA immunoprecipitation (RIP) assays using an antibody against endogenous human TAF10 on HeLa cell polysome extracts. Potential target mRNAs were tested by RT–qPCR. Data points correspond to technical duplicates from $n = 3$ biological replicates. **c**, RIP-coupled RT–qPCR assays using an antibody against endogenous mouse TAF10, performed on mESCs. Data points represent technical duplicates from $n = 2$ biological replicates. **d**, Schematic representation of the GFP-RIP-coupled RT–qPCR assay using HeLa cell lines expressing doxycycline (Dox)-inducible GFP-TAFs to systematically probe co-translational assembly in TFIID. **e**, Matrix summarizing the results of the systematic GFP-RIP assay in **d**. GFP-tagged TFIID subunits were used as baits in a GFP-RIP assay from polysome extracts (rows), and enrichment for TFIID-subunit mRNAs was assessed by RT–qPCR (columns). The area of each circle is proportional to mRNA $\log_2$(fold enrichment (FE)) over mock IP. Combinations whose FE was less than fourfold of that of negative control target mRNA (*PPIB*) are not shown in the plot and are considered negative. Gray circles

represent hits for bait mRNA. Black circles represent Co-TA hits. Red circles highlight the widespread enrichment for *TAF1* mRNA from RIP of several TFIID subunits. Stars indicate subunits for which GFP fusion resulted in ambiguous protein functionality. Results represent the mean of $n = 2$ biological replicates. **f**, RIP-coupled RT–qPCR assays against endogenous TAF6 performed on HeLa cells. The C-terminal location of the epitope prevented the detection of the nascent TAF6 protein, along with its own mRNA, and the simultaneous co-TA with TAF9 (TAF6 HFD partner). Data points correspond to technical triplicates from $n = 2$ biological replicates. **g**, RIP-coupled RT–qPCR assays against endogenous TAF7. Data points correspond to technical triplicates from $n = 2$ biological replicates. Bar graphs in the figure show the mean of the data. Antigen regions for the antibodies used are indicated. HFD, histone-fold domain; HEAT, HEAT-repeat domain; TAF1iD, TAF1-interaction domain; FRT-TO, FLP Recombination Target-Tet-ON; $P_{tet}$, tetracycline-responsive promoter. Cycloheximide (CHX) prevents ribosome dissociation from the mRNA. By contrast, puromycin (Puro) induces premature nascent polypeptide chain termination and release from the ribosome or mRNA.

1% of *CTNNB1* mRNAs did so. The fraction of co-localized TAF1 mRNAs robustly decreased upon puromycin treatment, confirming the co-TA between the two subunits.

We then performed the IF experiment with TAF4 (part of core-TFIID), TAF7 (lobe C component) and TAF10 (lobes A and B component). All tested TAFs positively co-localized with *TAF1* mRNA (Fig. 3c–e).

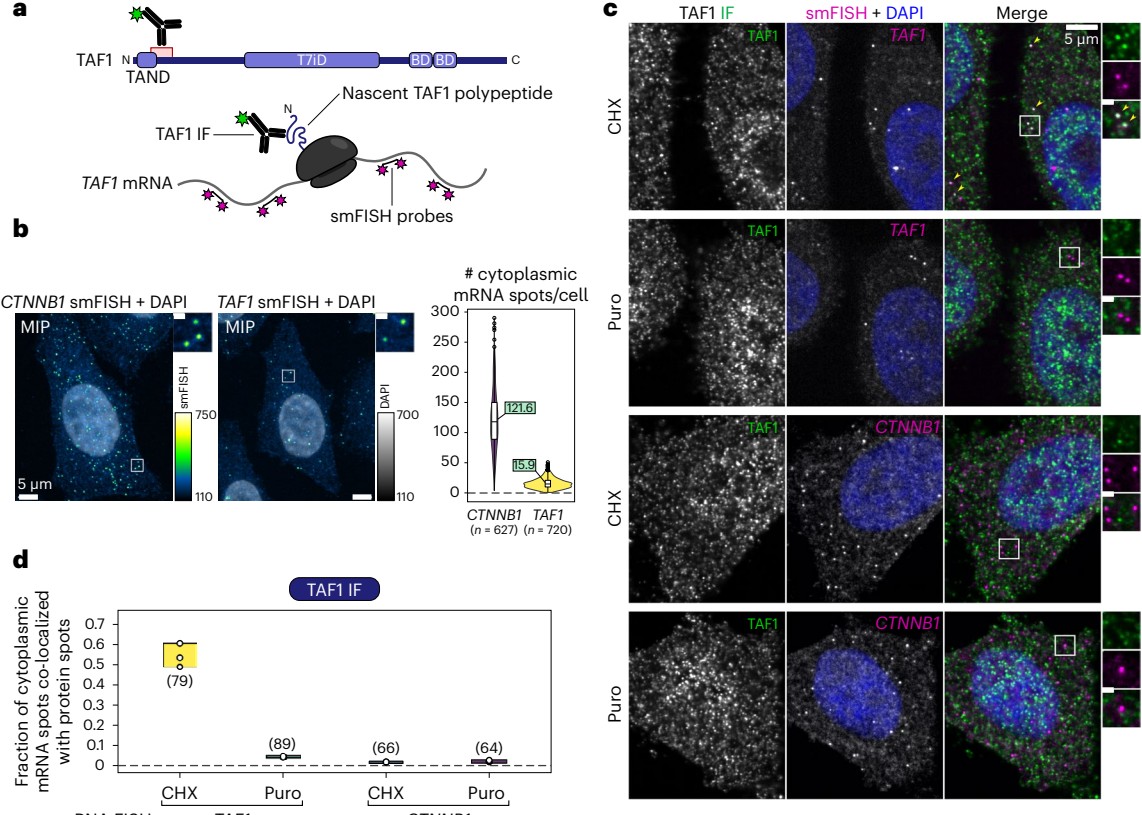

**Fig. 2 | Endogenous nascent TAF1 protein detection. a**, Schematic overview of the imaging strategy used to detect actively translated *TAF1* endogenous mRNAs with a combination of smFISH and IF. The antigen region recognized by the TAF1 antibody used in the assay is indicated. TAND, TAF1 N-terminal domain; T7iD, TAF7-interaction domain; BD, bromodomain. **b**, Representative confocal maximum intensity projections (MIPs) of smFISH against the negative control *CTNNB1* and *TAF1* mRNAs in HeLa cells. The plot on the right shows the absolute number of cytoplasmic mRNAs per cell (mean values are shown in green boxes, and total number of cells in brackets). In the boxplots, the center line marks the median, the box bounds mark the 25th and 75th percentiles, and the whisker limits

are 1.5 × interquartile range. **c**, Representative multicolor confocal images for the co-localization assay shown in **a** and quantified in **d**. Each image is a single confocal optical slice. TAF1 protein IF and *TAF1* mRNA detection in the merged image are shown in green and magenta, respectively. Co-localizing spots are indicated with yellow arrows. Zoomed-in regions (white squares) are shown on the right. Inset scale bars, 1 μm. **d**, Quantification of the fraction of mRNAs co-localized with protein signal for each experimental condition. Each open circle corresponds to an independent field of view (*n* = 3, total number of cells is in brackets).

TAF4 (Fig. 3c) and TAF10 (Fig. 3e) co-localization levels with *TAF1* were comparable to those of TBP (Fig. 3b), whereas levels for TAF7 (Fig. 3d) were considerably higher: ~40% of *TAF1* mRNAs co-localized with TAF7 spots. Puromycin treatment consistently reduced the fraction of co-localization for all TAFs, although the reduction for TAF4 was not as large. *TAF1* co-localization with SUPT7L—a subunit of the SAGA complex[27]—was very low (<1%) and not affected by puromycin (Fig. 3f), confirming the specificity of the results. We then probed cells for TBP and TAF7 subunits simultaneously using dual-color IF (Extended Data Fig. 3a). We found that ~50% of TBP-positive *TAF1* RNA spots were simultaneously co-localized with TAF7. Puromycin treatment drastically reduced the frequency of co-localization, nearly abolishing the double-positive events (Extended Data Fig. 3b). Overall, these observations support the data from our systematic RIP–qPCR studies, further suggesting that multiple TFIID subunits are recruited on the TAF1 nascent polypeptide during TAF1 protein synthesis.

## The cytoplasm is populated by multisubunit TFIID 'building blocks'

To better understand how the co-TA events that we described above may participate in TFIID assembly, we set out to analyze the composition of potential TFIID assemblies in the cytoplasm. To this end, we immunopurified endogenous TFIID subunits from HeLa cytoplasmic extracts and

analyzed the immunoprecipitated endogenous complexes by label-free mass spectrometry (MS) (Fig. 4a–f). All of our immunoprecipitation assays (IPs) invariably retrieved holo-TFIID from nuclear extracts, confirming the effectiveness of the antibodies used (Extended Data Fig. 4a).

Cytoplasmic TAF2 was found associated with TAF8, in accordance with their co-TA (Fig. 1e), and TAF2 was partially integrated in the 8TAF complex (composed of core-TFIID, TAF2, TAF8 and TAF10; Fig. 4)[21,28]. The majority of immunopurified cytoplasmic TAF4 was in complex with TAF8, TAF6, TAF9/9B, TAF10 and TAF5 (Fig. 4b), which we interpreted as the 7TAF complex (core-TFIID, TAF8 and TAF10; the missing detection of TAF12 could be owing to the documented post-translational modifications of this subunit). The presence of small amounts of TAF11 co-purified with TAF4 hinted at the incorporation of the latter in a partial A lobe. In the cytoplasmic anti-TAF4 IPs, we found only TAF4, while in the nuclear anti-TAF4 IP we found peptides from TAF4 and its paralog TAF4B (Fig. 4b and Extended Data Fig. 4a). These findings suggest that the isolated cytoplasmic TAF4-containing building block is either lobe A or lobe B (as indicated in Fig. 4g), containing only one copy of TAF4. By contrast, the detection of both TAF4 and TAF4B in nuclear TAF4 IP suggests the isolation of holo-TFIID, as the holo-complex contains two copies of TAF4 family members (TAF4 and TAF4B).

Endogenous cytoplasmic TAF10 IP retrieved similar amounts of TAF10's HFD partner TAF8, with a relevant portion of the heterodimer

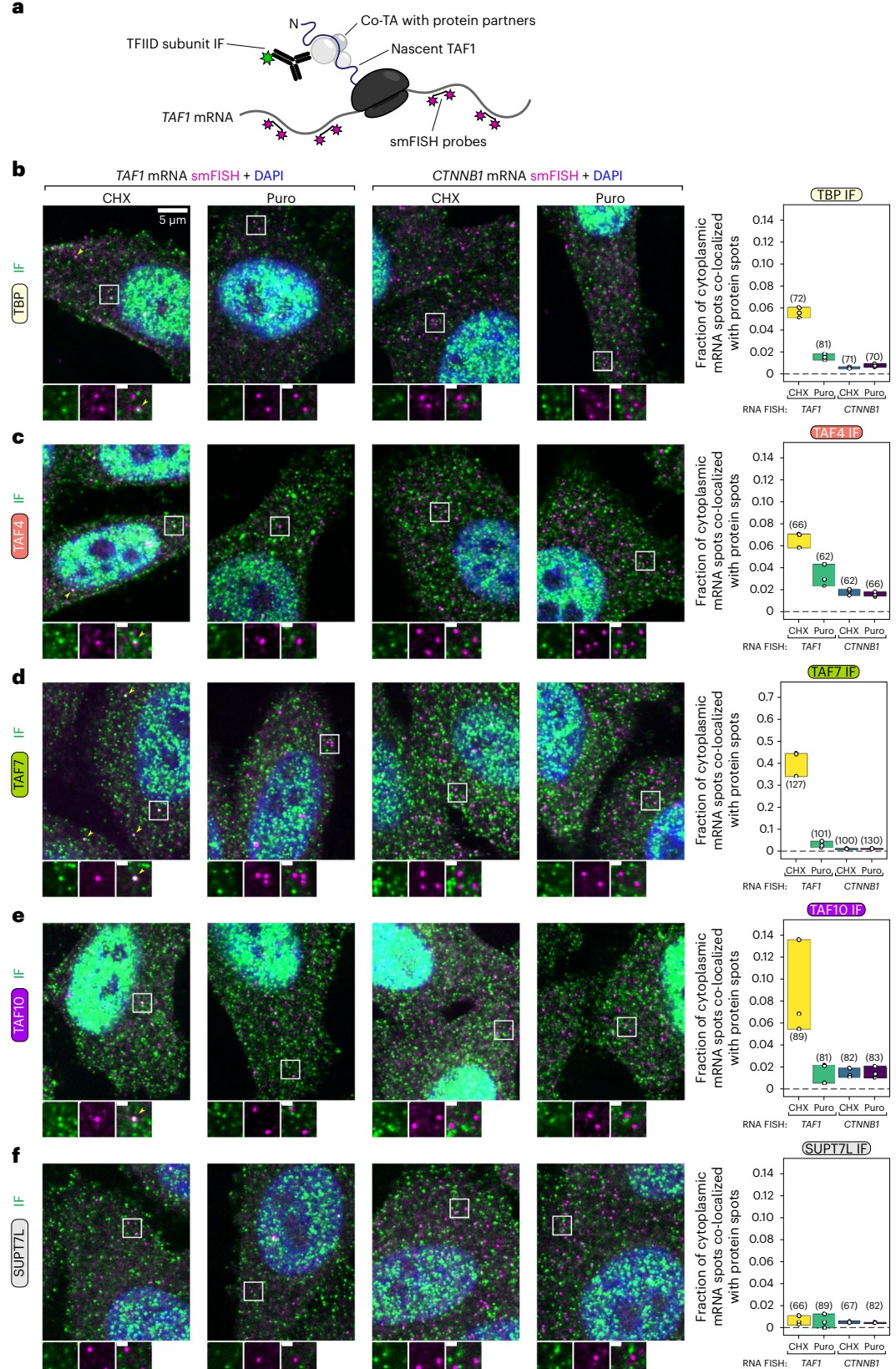

**Fig. 3 | Endogenous TFIID subunits are localized in physical proximity to *TAF1* mRNA in the cytoplasm of human cells. a**, Schematic overview of the imaging strategy used to detect co-TA events of endogenous TFIID subunits on *TAF1* mRNA with a combination of smFISH and IF. **b–f**, Representative multicolor confocal images from the co-localization assay shown in **a** for each assessed subunit. Each image is a single multichannel confocal optical slice. Protein IF and *TAF1* mRNA detection are shown in green and magenta, respectively. Co-localizing spots are indicated with yellow arrows. Zoomed-in regions (white squares) are shown below each image. Inset scale bars, 1 μm. The plots on the right report the fraction of target mRNAs co-localized with protein signal for each experimental condition. Each open circle corresponds to an independent field of view (*n* = 3, total number of cells is in brackets).

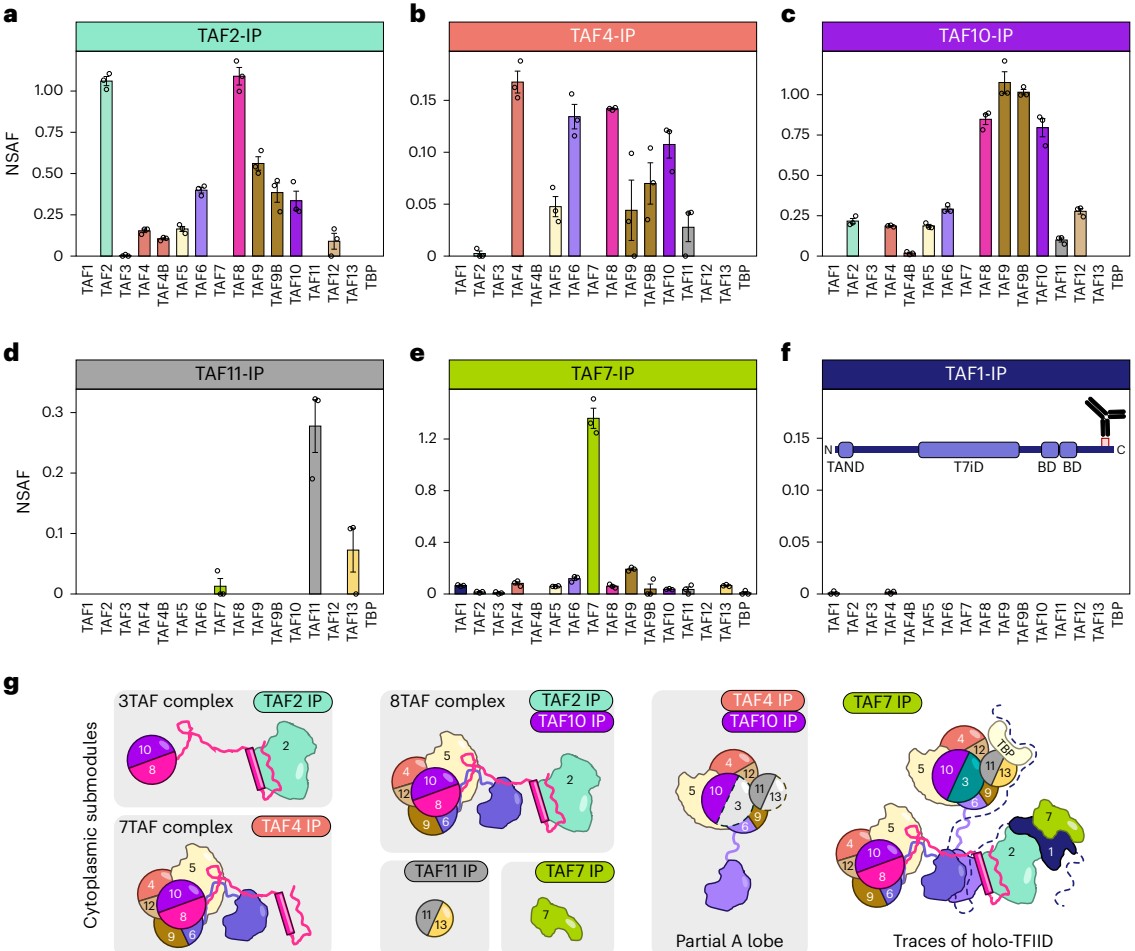

**Fig. 4 | The cytoplasm is populated by multisubunit TFIID 'building blocks'.**
**a–f**, IP of endogenous TFIID subunits coupled to label-free MS, performed on cytoplasmic extracts from human HeLa cells. Bar plots represent the average NSAF (normalized spectral abundance factor) value for each detected subunit in technical triplicates. Error bars represent the s.e.m. The antigen position of the TAF1 antibody used for IP is shown in **f. g**, Visual summary of cytoplasmic TFIID submodules inferred from IP–MS data. IPs in which the given submodule was enriched are indicated.

associated with core-TFIID and TAF2 in the 8TAF complex (Fig. 4c). In this case, the small amounts of TAF11 also hint that a fraction of TAF10 is incorporated in a partially assembled A lobe. On the other hand, we found some TAF11 associated with its HFD-partner TAF13 in the TAF11 IP[20], with no detectable amounts of other lobe A subunits (Fig. 4d). Most immunopurified TAF7, the partner of TAF1 in TFIID, was not in the complex (Fig. 4e). Yet, cytoplasmic TAF7 co-purified with trace amounts of TFIID subunits—including TAF1 (Extended Data Fig. 4b). In contrast to nuclear extracts (Extended Data Fig. 4a), cytoplasmic TAF1 IP did not enrich any TFIID component, with the bait itself being barely detectable (Fig. 4f). Given that the IP-grade TAF1 antibody recognizes a C-terminal epitope along the protein, we conclude that the abundance of TAF1 mature protein in the cytoplasm is below the detection limit in this analysis.

These results demonstrate that the cytoplasm of HeLa cells is populated by different multisubunit TFIID submodules, likely representing stable intermediates along the assembly pathway of the complex (Fig. 4g). None of the cytoplasmic IPs, except for TAF7, co-purified TAF1, suggesting that it is present in small amounts in the cytoplasm and is the limiting factor in TFIID assembly. These findings further point to a co-translational recruitment mechanism whereby the preassembled TFIID 'building blocks' associate with nascent TAF1 polypeptide, in agreement with our RIP and imaging experiments.

## TAF1 crosslinking hotspots are anchor points for TFIID building blocks

TAF1 is the largest subunit of TFIID (1,872 amino acids), and only ~47% of the protein structure has been solved. To rationalize how the nascent TAF1 polypeptide could work as a hub for TFIID assembly, we analyzed all available crosslinking-MS experiments performed on highly purified TFIID or PIC-incorporated TFIID[10,18,19]. The intercrosslinks between TAF1 and other TFIID subunits detected in at least two independent datasets indicate three main proximity and crosslinking 'hotspots' along TAF1 (Fig. 5a and Supplementary Table 2): (1) a loose region crosslinked with TBP and its interacting partners TAF11 and TAF13; (2) a well-defined hotspot rich in crosslinks with TAF6 along with single positions associated with TAF5, TAF8 and TAF9; and (3) a large central region that was extensively crosslinked to TAF7 and, to a lesser extent, to TAF2. The combination of the crosslinking hotspots (Fig. 5a) with TAF1 sequence features (conservation and structural disorder; Fig. 5b), annotated functional domains (Fig. 5c) and structural observations (Fig. 5d) shows that TAF1 is a flexible scaffold protein that connects all TFIID submodules by three main anchor points.

TAF1 modular organization is shown on the AlphaFold model of the full-length protein (Fig. 5c). A substantial fraction of the protein (~48%) is predicted to be intrinsically disordered, including interdomain linker regions and the long acidic C-terminal tail (Fig. 5b). TAF1 contains two main well-structured regions: the TAF7 interaction domain

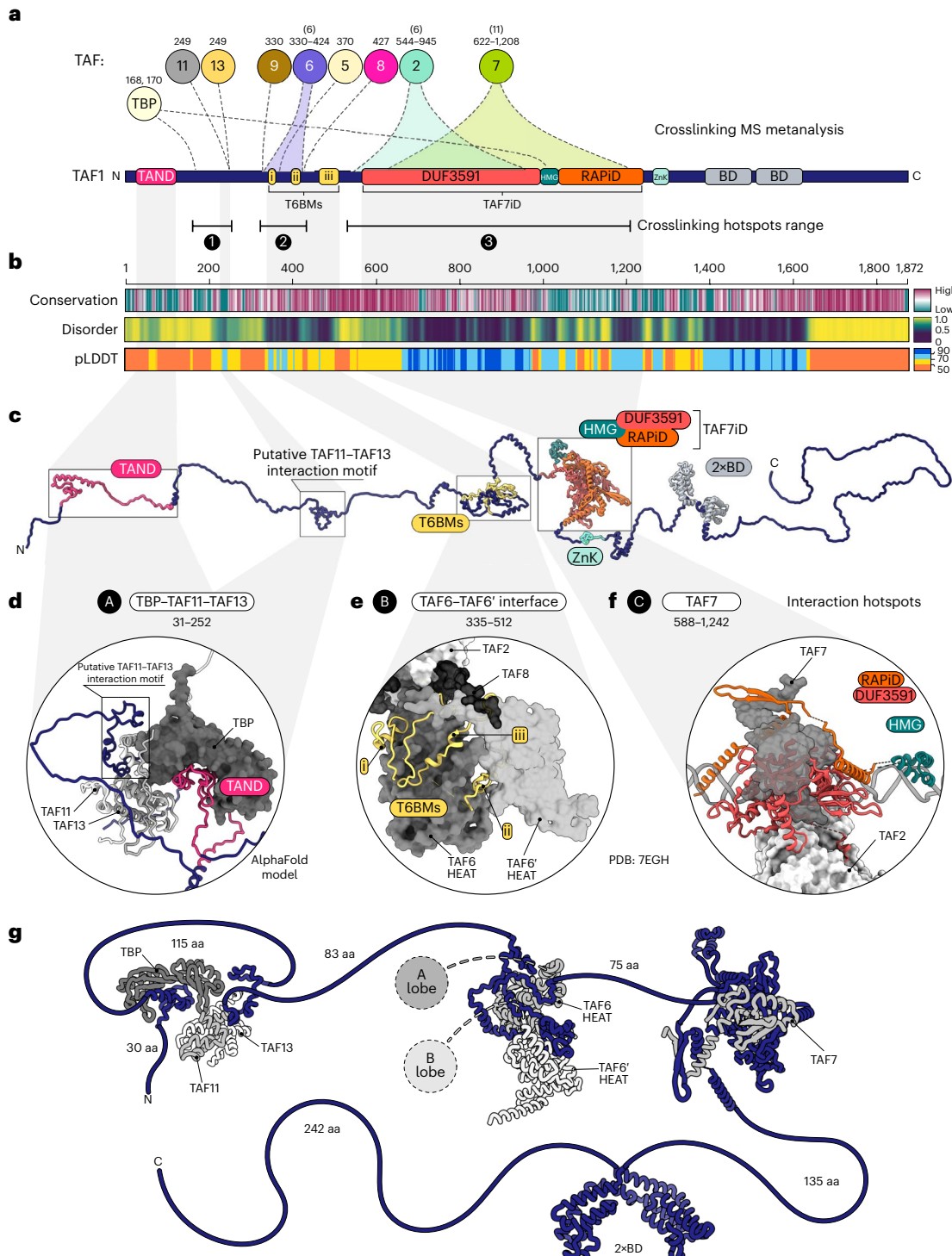

**Fig. 5 | Three crosslinking hotspots identified on TAF1 correspond to distinct anchor points for specific TFIID building blocks. a**, Summary of TAF1-centred crosslinking-MS metanalysis derived from three independent studies[10,18,19]. Interprotein crosslinks between TAF1 and other TFIID subunits (TAFs are indicated with their corresponding numbers) are displayed along the protein as dotted lines or shaded ranges. Numbers above each crosslinked subunit indicate TAF1 positions or the range involved in the crosslinks, and the values in brackets indicate the number of distinct TAF1 crosslinked positions. Only crosslinks reported in at least two independent datasets are shown. The known structural domains of TAF1 are depicted. **b**, Heatmaps of TAF1 conservation (ConSurf), structural disorder (Metapredict) and pLDDT (AlphaFold structural prediction confidence) scores are shown. Structures with pLDDT > 90 are expected to be modeled to high accuracy. A pLDDT between 70 and 90 means that the structure

is modeled well, and structures with a pLDDT between 50 and 70 should be treated with caution. The scales of the different prediction scores are shown on the right. **c**, Full-length human TAF1 AlphaFold model. The initial structure was extended to better appreciate the different domains indicated along the protein. **d**–**f**, Structural models of the three main TAF1 anchor points (labeled A, B and C) in TFIID are shown in the insets. The model in **d** is the result of AlphaFold prediction of the TAF1–TAF11–TAF13–TBP subcomplex. Distinct TAF1 domains are colored as in **a**. Partner subunits are shown in shades of gray. TAF1 segments part of each anchor point are indicated. **g**, Schematic summary of the distinct interaction hotspots along the protein and the length of the intervening linker regions. TFIID lobes A and B are shown as circles. aa, amino acids; HMG: HMG-box domain; ZnK: zinc-knuckle domain.

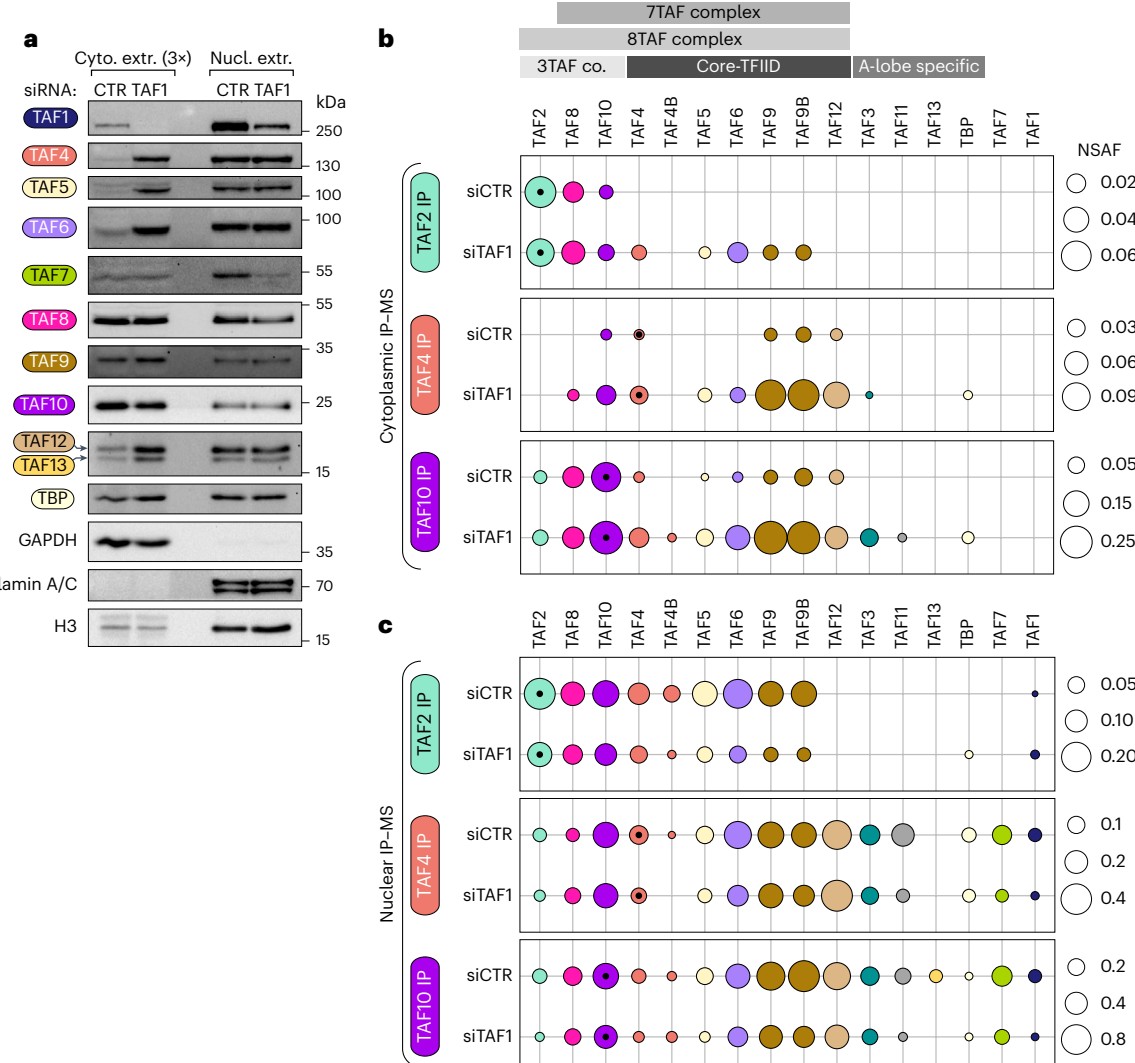

**Fig. 6 | TAF1 depletion leads to an accumulation of TFIID building blocks in the cytoplasm. a**, Subcellular fractionation of siRNA-transfected HeLa cells followed by western blot analysis of endogenous TFIID subunits distribution. GAPDH, lamin A/C and histone H3 were used as loading controls. The amount of loaded cytoplasmic (cyto.) extract is three times the amount of the nuclear (nucl.) extract counterpart. The positions of the molecular weight markers in kDa are indicated on the right. **b**, IP of endogenous TFIID subunits coupled to label-free mass-spectrometry (MS) performed on cytoplasmic extracts of siRNA-transfected HeLa cells. The circle area represents the average NSAF value for each detected subunit in technical triplicates. The NSAF scales are indicated on the right. Distinct TFIID subcomplexes are depicted on the top. Subunits are color-coded and arranged according to the subcomplexes indicated on the top. Black dots in the circles identify the protein used as bait in each IP. **c**, Same as in **b**, but the IPs were performed on nuclear extracts. CTR, non-targeting control siRNA. Co., complex.

(TAF7iD), which occupies the central portion of the protein, and two histone-reader bromodomains (BD) localized in tandem along the C-terminal tail (Fig. 5a). The TAF7iD, composed of the DUF3591 domain in concert with the RAP74 interaction domain (RAPiD), tightly associates with TAF7 and binds downstream of core promoter DNA[18,29]. Accordingly, these regions were modeled with high confidence by AlphaFold (Fig. 5b). The scarcity of TAF1 intraprotein crosslinks outside the TAF7iD and the tandem BDs is in accordance with the absence of other major structured domains along the protein (Extended Data Fig. 5a).

The three described crosslinking hotspots correspond to distinct anchor points (named here A, B and C) for specific TFIID submodules (Fig. 5d–f). The first TAF1 anchor point (A) would interact with TBP and the TAF11–TAF13 heterodimer, as the flexible TAF N-terminal domain (TAND) has been shown to directly interact with TBP and inhibit TBP DNA binding[30–32]. Additionally, removal of human TAND has been found to abolish the co-translational recruitment of TBP to TAF1 (ref. 24). The crosslinks of the TAF11–TAF13 heterodimer with TAF1 are consistently

found in all datasets. They map on TAF1 Lys249, which lies within a conserved motif predicted with higher confidence and lower disorder scores than those of the flanking regions (Fig. 5a–c). Modeling TAF1 with TBP and TAF11–TAF13 with AlphaFold resulted in a ternary complex with the expected positioning of TAF1 TAND into the concave surface of TBP. The putative TAF11–TAF13 interaction motif of TAF1 was folded laterally in a pocket formed by the HFD subunits (Fig. 5d). The TAF1 Lys249 position in the model is compatible with all the experimental crosslinks with TAF11–TAF13 (Extended Data Fig. 5b).

The second hotspot (B) is the anchor point of both lobes A and B with TAF1. It is composed of three TAF1 stretches of conserved amino acids interspersed by loops of lower conservation, named TAF6-binding motifs (T6BMs, Fig. 5a–c). These motifs, which have recently been resolved by cryo-EM[10], bridge the two copies of TAF6 HEAT domains, which in turn are connected to lobes A and B (Fig. 5e, see also Fig. 1a). The T6BMs occupy defined grooves and pockets across the pair of TAF6 HEAT domains at the center of TFIID (Fig. 5e). Modeling the entire

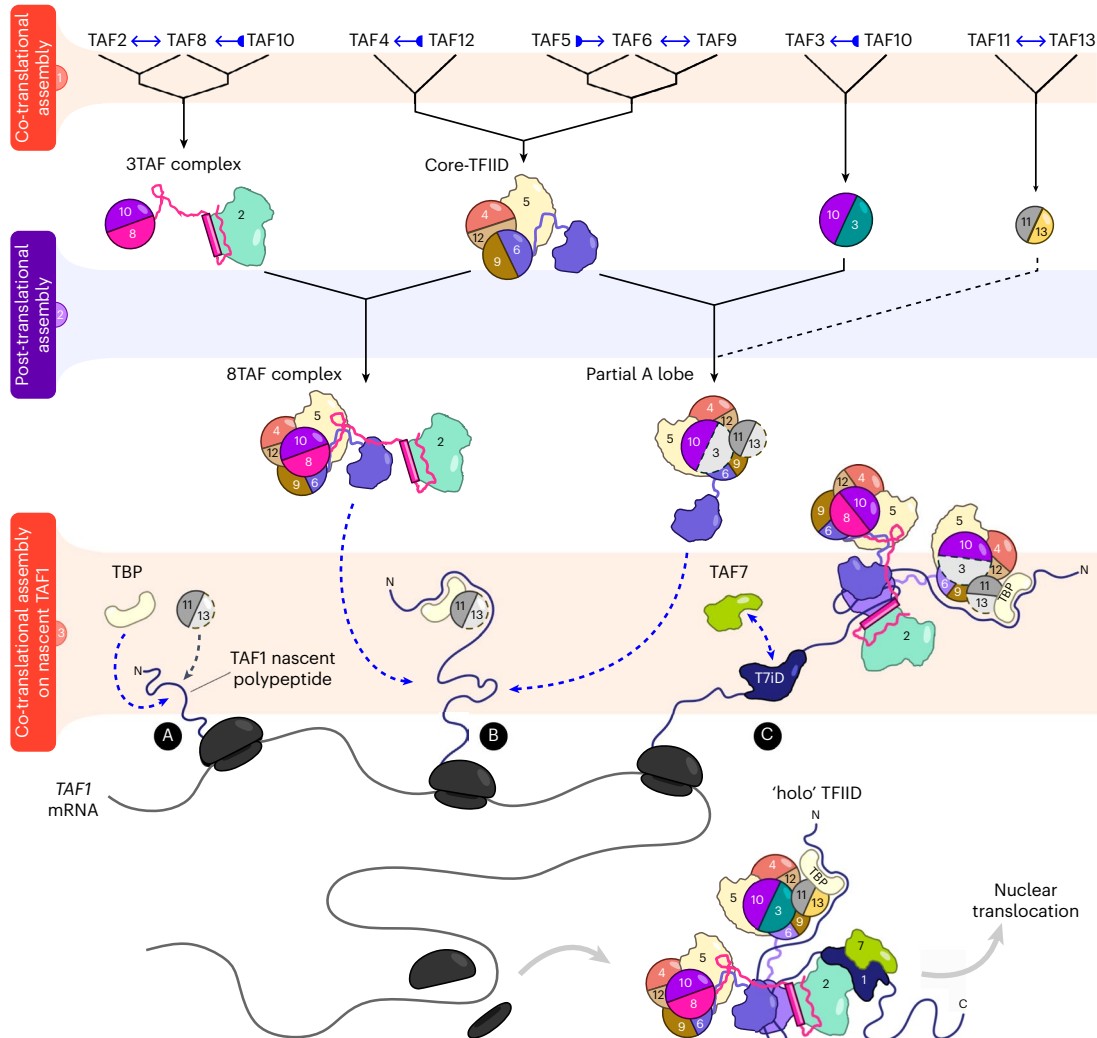

**Fig. 7 | A co-translational hierarchical model for TFIID assembly.** Scheme of the proposed cytoplasmic assembly model for TFIID that reconciles the experimental observations of the present work with previous structural and biochemical data. The assembly pathway can be subdivided into three tiers (colored and numbered horizontal stripes). Tier 3 represents the co-translational assembly of several TFIID building blocks on nascent TAF1 protein through three distinct interaction hotspots (labeled A, B and C), resulting in TFIID. Blue arrows with bases indicate directional co-TA events, and double-headed blue arrows specify reciprocal co-TA, as assessed by RIPs. For further details, see the Discussion section.

TAF1 region containing the T6BMs allowed us to map all crosslinking sites otherwise positioned in unresolved flexible loops (Extended Data Fig. 5c), in perfect agreement with the experimental structure (Extended Data Fig. 5d–f). Overall, the T6BMs constitute most of the interface anchoring the two copies of TAF6 HEAT domains together (Extended Data Fig. 5g,h). The absence of crosslinked positions along the third T6BM (Fig. 5a) is due to the lack of lysine residues. Apart from TAF6, the crosslinks to other TAFs within this hotspot are likely driven by proximity rather than direct interactions.

The third hotspot (C) coincides with the TAF7iD (Fig. 5f). Besides the intricate fold adopted with TAF7, the DUF3591 loosely anchors the resulting TAF1–TAF7 globular domain to TAF2 (Extended Data Fig. 5i). Overall, structural and biochemical data support a scaffolding function of TAF1 within TFIID, thanks to its modular organization (Fig. 5g). TAF1 represents a flexible three-way anchor point that physically connects the three TFIID lobes through direct interactions with the two copies of TAF6, which in turn emanate into lobes A and B (Fig. 5g). Strikingly, all mapped crosslinks reside in the N-terminal half of TAF1 (Fig. 5a), leaving the ~700-aa region downstream of RAPiD free from crosslinks. This would allow TFIID assembly on the N-terminal half of TAF1 before the protein is released from the ribosome.

## TAF1 depletion leads to an accumulation of TFIID building blocks in the cytoplasm

To investigate the role of TAF1 in the dynamics of cytoplasmic TFIID assembly, we perturbed the TAF1-dependent assembly by siRNA-mediated TAF1 knockdown (KD). Subcellular fractionation experiments revealed an enrichment of the protein levels of several TFIID subunits in the cytoplasmic fraction upon TAF1 KD. Specifically, the cytoplasmic extract was substantially enriched for core-TFIID subunits (TAF4, TAF5, TAF6 and TAF12) (Fig. 6a). This cytoplasmic increase in protein levels of TAF4/5/6/12 was not visible in the nuclear fraction, suggesting a specific cytoplasmic accumulation of those subunits. Also, TAF13, and to a lesser extent TBP, followed the same pattern. Instead, the levels of TAF8 and its partner TAF10 remained mostly unchanged. Notably, although the levels of TAF7 stayed constant in the cytoplasm, they were drastically reduced in the nuclear fraction, closely matching the depletion of TAF1. These observations show that TFIID subunits are differentially affected by TAF1 depletion. On the contrary, TAF4 and TAF7 KD under the same conditions did not reproduce the effect elicited by TAF1 silencing, suggesting that the observed phenomenon is TAF1-specific (Extended Data Fig. 6a).

To address whether the cytoplasmic increase of a subset of TFIID subunits would correspond to an accumulation of specific TFIID building blocks in the cytoplasm, we analyzed endogenous cytoplasmic subcomplexes composition by IP–MS upon TAF1 KD (Fig. 6b and Extended Data Fig. 6b). We selected IP-grade antibodies raised against a lobe C subunit (TAF2), a core TFIID subunit (TAF4) and a non-core TFIID subunit (TAF10). In good agreement with our findings, upon TAF1 KD, TFIID building blocks accumulated in the cytoplasm. The enrichment of cytoplasmic core-TFIID was evidenced in all IPs. Notably, the lobe A-specific subunits, TAF3, TAF11 and TBP, co-purified with TAF4 and TAF10 only following TAF1 depletion. We interpret these results as the cytoplasmic accumulation of different TFIID building blocks, including the 8TAF complex (B lobe+TAF2) and the A lobe complex, provoked by the impairment of the last, TAF1-dependent, assembly step before nuclear import.

IP–MS analyses on the nuclear fraction showed less dramatic rearrangements in subunit distribution, with an overall decrease in the abundance of all immunopurified TFIID subunits in TAF1 KD samples (Fig. 6c and Extended Data Fig. 6c). Overall, our observations together show that TAF1 is a major hub for the co-translational assembly of TFIID complex, from preassembled building blocks to subsequent nuclear translocation.

## Discussion

The self-assembly of large heterotypic multiprotein complexes in living cells poses major challenges to our understanding of cellular homeostasis. Here, we tackled the longstanding question of where and how the basal transcription factor TFIID assembles, and we comprehensively explored the landscape of co-TA events within TFIID. We uncovered TAF1 as the central hub in the assembly process and several previously undiscovered pairs of subunits that undergo co-TA (see Supplementary Table 1).

### A hierarchical co-translational model for TFIID assembly

All our findings can be rationalized in a hierarchical model for TFIID assembly, which is stratified in three tiers of assembly events (Fig. 7). The first tier includes early events along the pathway: these are the formation of protein pairs, mostly through the dimerization of HFD-containing subunits. We find it remarkable that all the HFD pairs in TFIID assemble co-translationally, either directionally or symmetrically. The fact that several subunits used as bait in our cytoplasmic IP–MS data were not found as free proteins (Fig. 4) points either at a fast and efficient co-TA with their partners or to a degradation-driven removal of orphan subunits, although a combination of the two processes is also likely. Tier 1 also harbors interactions of non-HFD subunits, such as TAF2 and TAF5, which interact co-translationally with TAF8 and TAF6, respectively. All these directly interacting pairs are structurally well characterized[10,18]. The products of tier 1 assembly are free early multisubunit intermediates, likely stabilized by interactions with their partner. They are likely characterized by heterogeneous half-lives as free molecular species, since some of them can be isolated in our steady-state experiments (for example, TAF11/TAF13 HF pair), whereas others can be detected only as part of larger assemblies (for example, TAF4/TAF12 HF pair), yet some others are not detected at all (for example, TAF3/TAF10 HF pair) (Fig. 4). The products of tier 1 in turn access the second level of the assembly pathway by combining with each other in a few structurally constrained steps. Assembly in tier 2 occurs post-translationally and leads to the buildup of larger assemblies that were recurrently found in our IP–MS experiments, such as the 8TAF complex and a partially assembled lobe A.

In tier 3, the products of tier 2 finally converge and engage co-translationally with the nascent TAF1 polypeptide (Fig. 7). An appealing idea is a sequential N- to C-terminal order of assembly, whereby different TFIID building blocks are recruited by the distinct assembly domains of nascent TAF1 as they emerge from the ribosome channel. The first N-terminal anchor point (A) would interact with TBP, which

engages with nascent TAF1 by binding the TAND domain[24]. Our systematic survey confirmed this co-TA pair. TAF11–TAF13 dimer could also engage with TAF1 at anchor point (A), forming a ternary complex along with TBP (Fig. 5d). Biochemically, a recombinant complex formed by TAF1–TBP–TAF11–TAF13 and TAF7 can be readily purified[33], and the direct interaction between TAF1 and TAF11–TAF13 is supported by crosslinking experiments and structural modeling (Fig. 5c,d and Extended Data Fig. 5b). However, TAF11–TAF13 did not score positive for *TAF1* mRNA in our systematic RIP approach, opening the possibility of post-translational engagement, or weaker interactions.

The second interaction anchor point (B)—the T6BMs—would interact with two copies of TAF6 HEAT domains, bringing together lobe A and lobe B (Figs. 7 and 5g). Interestingly, TAF1 evolved distinct binding motifs to recognize corresponding identical surfaces from the two TAF6 copies within TFIID (Extended Data Fig. 5h). The third anchor point (C) recruits TAF7, which interacts with the TAF1 central domain (DUF3591 and RAPiD) (Fig. 5g). Notably, in our RIP experiments, TAF7 enriched *TAF1* mRNA and vice versa, opening the possibility of a simultaneous co-translational interaction between the two. Such an ordered addition would entail a remarkable degree of coordination, potentially reinforced by binding cooperativity among the modules as they join the growing assembly. Yet, in our imaging data, we detected TAF7-positive *TAF1* RNA spots lacking TBP signal and vice versa, hinting at a potential independent binding mode (Extended Data Fig. 3).

Upon completion of TAF1 protein synthesis, the assembled TFIID is released and readily translocated in the nucleus. A subset of subunits scored negative for *TAF1* mRNA in our RIP assays: these include TAF3, TAF9, TAF11 and TAF13. Therefore, it is possible that they join the complex post-translationally or through their interaction partners. The benefits of a hierarchical co-translational assembly have been recently theorized in the framework of yeast nuclear pore assembly[9]. The proposed model may also apply to our findings, in which co-TA is pervasively exploited for the hierarchical assembly of TFIID in the cytoplasm of mammalian cells.

Our data are in agreement with the published TAF interactions, the cryo-EM TFIID structures[10,23] and the previous descriptions of partial TFIID assemblies[28,34]. According to recent bioinformatic analyses, during evolution, proteins that assemble co-translationally have sustained large N-terminal interfaces in order to promote co-translational subunit recruitment[35]. In agreement, out of the eight larger subunits of TFIID that participate in co-TA as nascent polypeptides (TAF1, TAF2, TAF3, TAF4, TAF6, TAF7, TAF8 and TAF9), all except TAF4 have their interaction domains in the N terminus.

### TAF1: a 'driver' and limiting factor along the assembly line

*TAF1* mRNA was enriched in the majority of our RIPs (Fig. 1 and Supplementary Table 1), and it was found in physical proximity of several TFIID subunits in the cytoplasm (Fig. 3). In this compartment, the levels of TAF1 protein seem to be limiting with respect to TFIID building blocks (Fig. 4). TAF7 IP enriched the whole spectrum of TFIID subunits, including TAF1, albeit at very low levels (Fig. 4e and Extended Data Fig. 4b). TAF7 also showed the highest levels of co-localization with *TAF1* mRNA (Fig. 3d), pointing at a remarkable co-TA efficiency between TAF1 and TAF7. The higher assembly efficiency is consistent with the detection of a fully assembled complex in TAF7 cytoplasmic IP. The interaction interface between TAF1 and TAF7 is remarkably intricate, with deeply intertwined β-strands from each protein contributing to a common β-barrel[29]. It would be conceivable that such an interface would form only concomitantly with folding during protein synthesis, imposing a structural constraint solved by co-TA. Curiously, TAF7 levels decreased proportionally with TAF1 depletion in the nucleus, hinting at a partner stabilization effect (Fig. 6a), similar to the one observed between TAF10 and TAF8 (ref. 24).

TAF1 depletion led to the accumulation of several TFIID building blocks in the cytoplasm, revealing a key role of this subunit in driving

complex assembly and consequent relocation in the nucleus (Fig. 6). We propose that nascent TAF1 nucleates the late steps of TFIID assembly in the cytoplasm by tethering different submodules of the complex, and, once released from the ribosomes, the whole assembly efficiently shuttles in the nucleus (Fig. 7). This process may act as a quality checkpoint before nuclear import. In agreement, both in yeast and in metazoans *TAF1* is an essential gene[36–38].

A central role of a single nascent subunit for the co-translational assembly of protein complexes has been demonstrated for the COMPASS histone methyltransferase in yeast, in which a specific subcomplex is directly assembled on nascent Set1 protein, stabilizing the latter from degradation[39]. Set1 behaved as a co-translational 'driver' subunit, simultaneously promoting complex assembly and limiting its abundance. Other examples of subunits potentially working as co-translational drivers for complex assemblies have been uncovered in fission yeast[4] and are supported by structural analyses[40]. We argue that an equivalent process in mammalian cells is led by TAF1 as the driver subunit for TFIID co-translational assembly, which culminates with the tethering of distinct building blocks on TAF1 nascent polypeptide. In this regard, *TAF1* mRNA offers the longest coding sequence (CDS) among TFIID components, implying there is a prolonged timeframe in which co-translational binding events can occur. By taking into account an estimate of average translation speed of ~5.6 codons s$^{-1}$ in mammalian cells[41], translating *TAF1* CDS would take ~5.6 min. The last assembly domain along TAF1 completely emerges from the ribosome around position 1240, granting an additional window of time of about 1.9 min to ultimate co-TA before ribosome release. In this regard, the analysis of ribosome profiling (Ribo-seq) merged datasets showed a wide region of sparse ribosome-protected fragments, encompassing all three T6BMs and extending inside the DUF3591 domain-encoding region (Extended Data Fig. 7). The low signal in the T6BMs region hints at fast elongation rates, which would rapidly expose all three T6BMs for the co-translational recruitment of the respective TFIID building blocks. Downstream of this region, translation slows down, as suggested by the higher ribosome occupancy. This would also buy time to establish productive co-translational interactions with upstream anchor points.

## Open questions

Our findings reveal an unprecedented mechanism for TFIID biogenesis, answering longstanding questions and opening new ones. One of remarkable importance is how efficiently co-TA occurs. A prerequisite for co-translational interactions is an actively translated mRNA. Our imaging approach detected nascent TAF1 protein on roughly half of the correspondent cytosolic messengers (Fig. 2). By using this observation as a proxy for the proportion of actively translated *TAF1* mRNAs, the observed frequency of Co-TA events for the other probed subunits (Fig. 3) would be underestimated. In the future, the observation of co-translational binding events in living cells might offer a quantitative dimension to this field.

A second point is whether co-TA is an efficient option for complex assembly or an obligate path. Co-TA might be the sole opportunity for assembly domains characterized by structural constraints, such as the TAF1–TAF7 interface. Instead, interactions mediated by classical binding pockets, extended surfaces or short linear motifs can rely also on post-translational assembly. However, co-translational interactions have the advantage of abolishing partially unfolded/unstable intermediates by kinetically anticipating their complexed state. Although it seems reasonable to hypothesize that natural selection promoted molecular features favoring co-translational interactions, it has proven hard to disentangle co-translational from post-translational assembly experimentally, since both mechanisms ultimately depend on protein synthesis. Note also that other chaperone-mediated assembly processes play a role in multisubunit complex assembly pathways[42].

Third, our data open new questions on the nuclear import mechanism adopted by TFIID or its building blocks. The observation that a defined set of subcomplexes accumulates in the cytoplasm upon TAF1 depletion opens the possibility of distinct entry routes to the nucleus. A fully assembled TAF1-containing complex could be the most efficiently translocated molecular species, with several subcomplexes relying on TAF1 for nuclear import. Conversely, other building blocks might access the nucleus autonomously, as shown for the TAF2–TAF8–TAF10 module[21,43]. The interrogation of a systematic interactome survey of the major nuclear transport receptors on human cells by BioID[44] showed that all the detected TFIID subunits shared the same import systems, mainly the α-importins IMA1 and IMA5. This is consistent with the idea that TFIID is transported across the nuclear pore as a pre-assembled entity. Intriguingly, in this study, TAF1 was one of the main biotinylated TFIID subunits, suggesting that TAF1 can directly interact with the nuclear transport receptors and drive nuclear import, as recently found in yeast[45].

Our study provides an understanding of a series of steps underlying the assembly mechanism of the general transcription factor TFIID. We envision that the principles of hierarchical co-translational assembly could apply to the biogenesis of most large heteromeric multiprotein complexes in living cells.

## Online content

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

## Methods

### Cell culture

Human HeLa cells (CCL-2; ATCC) were obtained from the IGBMC cell culture facility and cultured in DMEM (4.5 g L$^{-1}$ glucose) supplemented with 10% fetal calf serum (Dutscher, S1810), 100 U ml penicillin and 100 µg ml$^{-1}$ streptomycin (Invitrogen, 15140-130). E14 mESCs (ES Parental cell line E14Tg2a.4, Mutant Mouse Resource and Research Center) were obtained from the IGBMC cell culture facility and cultured on gelatinized plates in feeder-free conditions in KnockOut DMEM (Gibco) supplemented with 20 mM L-glutamine, penicillin–streptomycin, 100 µM non-essential amino acids, 100 µM β-mercaptoethanol, N-2 supplement, B-27 supplement, 1000 U ml$^{-1}$ LIF (Millipore), 15% ESQ FBS (Gibco) and 2i (3 µM CHIR99021, 1 µM PD0325901, Axon MedChem). Cells were grown at 37 °C in a humidified, 5% CO$_2$ incubator.

### GFP-fusion cell lines generation

GFP-TAFs fusion cell lines used in this study were described in ref. [26]. Briefly, the coding sequences for the human TFIID subunits (TAF1, TAF2, TAF3, TAF4, TAF5, TAF6, TAF7, TAF8, TAF9, TAF10, TAF11, TAF12, TAF13 and TBP) were obtained by PCR using the appropriate cDNA clone and gene-specific primers flanked by attB sites followed by BP-mediated GATEWAY recombination into pDONR221, according to the manufacturer's instructions (Invitrogen). The cloned sequence was verified by sequencing and it was transferred to the pcDNA5-FRT-TO-N-GFP Gateway destination vector by LR recombination, according to the manufacturer's protocol (Invitrogen). HeLa Flp-In/T-REx cells, which contain a single FRT site and express the Tet repressor[25], were grown in DMEM, 4.5 g L$^{-1}$ glucose (Gibco), supplemented with 10% vol/vol fetal calf serum (Gibco). All the GFP-fusion destination vectors were co-transfected with a pOG44 plasmid that encodes the Flp recombinase into HeLa Flp-In/T-REx cells using polyethyleneimine (PEI) to generate stable doxycycline-inducible expression cell lines. Recombined cells were selected with 5 µg ml$^{-1}$ blasticidin S (InvivoGen) and 250 µg ml$^{-1}$ hygromycin B (Roche Diagnostics) 48 h after PEI transfection. Cells were maintained in DMEM supplemented with 10% Tet-free fetal calf serum (Pan Biotech, P30-3602), blasticidin S, hygromycin B and penicillin–streptomycin.

### RNA immunoprecipitation against endogenous TFIID subunits

Polysome extract preparation and RIPs wexre performed essentially as described in ref. [24]. HeLa cells grown on 15-cm plates (~90% confluent) were treated either with 100 µg ml$^{-1}$ cycloheximide (CHX, Merck, C1988) for 15 min or with 50 µg ml$^{-1}$ puromycin (Puro, Invivogen, ant-pr-1) for 30 min in the incubator at 37 °C. Plates were placed on ice and cells were washed twice with ice-cold PBS and scraped in 2 ml lysis buffer (20 mM HEPES-KOH pH 7.5, 150 mM KCl, 10 mM MgCl$_2$, 0.1% NP-40, 1 × PIC (complete EDTA-free protease inhibitor cocktail, Roche, 11873580001), 0.5 mM DTT (ThermoScientific, R0862), 40 U ml$^{-1}$ RNasin Ribonuclease Inhibitor (Promega, N2511)) supplemented either with CHX or Puro. Cell suspension was homogenized with 10 Dounce strokes using a B-type pestle on ice. Lysates were incubated 15 min on ice and cleared by centrifugation at 17,000g. The supernatant represents the polysome extract.

For each IP, 1.2 mg protein G Dynabeads (Invitrogen, 10004D) was used. The antibodies used are listed in Supplementary Table 3. Dynabeads were washed twice in buffer IP100 (25 mM Tris-HCl pH 8.0, 100 mM KCl, 5 mM MgCl$_2$, 10% glycerol, 0.1% NP-40). Each antibody (5–10 µg per IP) was coupled to Dynabeads in 100 µL buffer IP100 for 1 h at room temperature (RT) while being agitated. Mock IPs were performed using mouse or rabbit IgG. Antibody-coupled Dynabeads were washed twice in buffer IP500 (25 mM Tris-HCl pH 8.0, 500 mM KCl, 5 mM MgCl$_2$, 10% glycerol, 0.1% NP-40) and three times in buffer IP100. For each IP, 1 ml of polysome extract (equivalent to ~10$^7$ cells) was used as input. A 10% equivalent volume of the input was kept at 4 °C for input normalization. IP reactions were incubated with rotation at 4 °C

overnight. The next day, Dynabeads were washed four times with 0.5 ml high-salt was buffer (25 mM HEPES-KOH pH 7.5, 350 mM KCl, 10 mM MgCl$_2$, 0.02% NP-40, 1 × PIC, 0.5 mM DTT, 40 U ml$^{-1}$ RNasin Ribonuclease Inhibitor) supplemented either with CHX or Puro. RNA from the resulting immunopurified material was extracted using NucleoSpin RNA XS kit (Macherey-Nagel, 740902) in 100 µL RA1 lysis buffer and purified according to the manufacturer's protocol. The input sample was extracted and processed in parallel with the IPs.

### GFP-fusion RIP

GFP-RIPs using inducible cell lines were performed as described for endogenous RIPs, with the following modifications. The day of the experiment, the expression of the GFP-tagged TFIID subunit was induced by addition of 1 µg ml$^{-1}$ doxycycline (Dox) for 2 h. For GFP-TAF3 and GFP-TAF10 cell lines Dox treatment was omitted due to their leaky basal expression. Cells were treated with CHX, lysed and polysome extracts prepared as described in the previous section. GFP-IPs were carried out using 40 µL GFP-Trap Agarose beads (ChromoTek, gta-20). Mock IPs were carried out using an equivalent volume of protein G Sepharose beads. Beads were incubated with polysome extracts for 4 h at 4 °C, washed and RNA purified as described in the previous section.

### RT−qPCR

Reverse transcription reaction was performed using SuperScript IV First-Strand Synthesis System (Invitrogen, 18091050) and random hexamers according to manufacturer instructions. The resulting cDNA was diluted 1:10. Two or three technical replicates of qPCR using 2 µL cDNA, primers listed in Supplementary Table 4 and LightCycler 480 SYBR Green I Master (Roche, 04887352001) were performed in a Light-Cycler 480 instrument (Roche). Input (%) normalization for RIP samples was performed by applying the formula $100 \times 2^{[(Ct_{input} - 6.644) - Ct_{RIP}]}$. Fold-enrichment normalization was performed by dividing RIP input (%) by mock input (%).

### Immunofluorescence–single-molecule inexpensive RNA FISH

RNA detection was performed using smiFISH[46]. Primary probe sets (24 single oligonucleotides) against target coding sequences were designed using Oligostan in R, as described in the software documentation[46]. Probes sequences are reported in Supplementary Table 4. Primary probes were synthesized by Integrated DNA Technologies (IDT) in plate format, and were dissolved in TE buffer at 100 µM. 5′ and 3′ Cy3-labeled secondary probe (FLAP) was synthesized by IDT and purified by high-performance liquid chromatography. Primary probes were mixed in an equimolar solution in TE at 0.83 µM per probe. To prepare a 50× smiFISH composite probes mix, 4 µL primary probe mix was mixed with 2 µL 100 µM secondary probe solution in 20 µL final reaction volume in 100 mM NaCl, 50 mM Tris-HCl pH 8.0 and 10 mM MgCl$_2$. The annealing reaction was performed in a thermocycler with the following conditions: 3 min at 85 °C, 3 min at 65 °C, 5 min at 25 °C. 50× smi-FISH probes mix was stored at −20 °C. The day before the experiment, HeLa cells were seeded on coverslips (no. 1.5H, Marienfeld, 630–2000) in a 12-well plate (0.2 × 10$^6$ cells per well). The day after, cells were treated either with 100 µg ml$^{-1}$ CHX for 15 min or with 50 µg ml$^{-1}$ Puro for 30 min in the incubator at 37 °C. Then, cells were directly processed for immunofluorescence. All buffer solutions were filtered (0.22-µm filter). Cells were washed twice with PBS (containing CHX for cells treated with it) and fixed with 4% paraformaldehyde (Electron Microscopy Sciences, 15710) in PBS for 10 min at RT. Cells were washed twice with PBS and incubated for 10 min at RT in blocking/permeabilization solution (BPS) (1× PBS, 1% BSA (MP, 160069), 0.1% Triton-X100 (Merck, T8787), 2 mM vanadyl ribonucleoside complexes (VRC, Merck, R3380)). Cells were incubated for 2 h at RT with the following primary antibodies diluted in BPS: TAF1 (1:1000, rabbit pAb, Abcam, ab188427), TAF4 (3 µg ml$^{-1}$, mouse mAb, 32TA 2B9), TAF7 (1:250, rabbit pAb, no. 3475), TAF10

(3 µg ml⁻¹, mouse mAb, 6TA 2B11), TBP (2 µg ml⁻¹, mouse mAb, 3TF13G3) or SUPT7L (rabbit pAb, Bethyl, A302-803A). A secondary-only control sample was incubated with BPS devoid of primary antibody. After three 5-min PBS washes, cells were incubated for 1 h at RT (light-protected) with AF488-conjugated secondary antibodies diluted 1:3,000 in BPS (goat anti-mouse IgG, A11001 or goat anti-rabbit IgG, A11008, Life Technologies). For dual-color IF, we also used Alexa Fluor Plus 647-conjugated secondary antibody (goat anti-mouse IgG, A32728). After three 5-min PBS washes, a second fixation step was performed with 4% paraformaldehyde in PBS for 10 min at RT. Cells were washed twice with PBS and equilibrated in hybridization buffer (2× SSC buffer, 10% formamide (Merck, F9037)) for at least 10 min at RT. An equivalent volume of the following mixes was prepared: Mix1 (2× smiFISH probes mix, 2× SSC buffer, 30% formamide, 0.68 mg ml⁻¹ *E. coli* tRNA (Roche, 10109541001)) and Mix2 (0.4 mg ml⁻¹ BSA (NEB, B9000S), 4 mM VRC, 21.6% dextran sulfate (Merck, D8906)). Mix1 and Mix2 were combined 1:1 and thoroughly mixed by vortexing. Then, 45 µL of the resulting solution were applied on the surface of a 10-cm plastic dish that served as hybridization chamber. Each coverslip was applied upside-down on the smiFISH mix drop. A hydration chamber (a 3.5-cm plate filled with hybridization buffer) was included. The hybridization chamber was sealed with parafilm and incubated overnight at 37 °C, in the dark. The day after, each coverslip was washed twice at 37 °C for 30 min in 2 ml hybridization buffer. In the second wash, 0.5 µg ml⁻¹ DAPI (Merck, MBD0015) was included for nuclear counterstain. After two PBS washes, coverslips were mounted with 5 µL Vectashield (Vector Laboratories, H-1000) and sealed with nail polish. For Alexa Fluor Plus 647 imaging, mounting was performed with Aqua-Poly/Mount[47] (Polysciences, 18606).

## Confocal microscopy and image processing

Cells processed for immunofluorescence/smFISH were imaged using spinning disk confocal microscopy on an inverted Leica DMi8 equipped with a CSU-W1 confocal scanner unit (Yokogawa), with a 1.4-NA ×63 oil-objective (HCX PL APO lambda blue) and an ORCA-Flash4.0 camera (Hamamatsu). DAPI, AF488 (IF) and Cy3 (smFISH) were excited using a 405 nm (20% laser power), 488 nm (70%) or 561 nm (70%) laser line, respectively. For dual-color IF experiments, Alexa Fluor Plus 647 was excited using the 642 nm laser line. Three-dimensional image acquisition was managed using MetaMorph software (Molecular Devices). Images of 2,048 × 2,048 pixels (16-bit) were acquired with a *xy* pixel size of 0.103 µm and a *z* step size of 0.3 µm (~30–40 optical slices). Multichannel acquisition was performed at each *z*-plane. Multicolor fluorescent beads (TetraSpeck Fluorescent Microspheres, Invitrogen, T14792) were imaged alongside the samples. Chromatic shift registration was performed with Chromagnon[48] using the fluorescent beads hyperstack as reference. Image channels were split, and maximum intensity projections (MIPs) were generated in Fiji[49] using a macro. smFISH RNA spots were detected and counted using the RS-FISH Fiji plugin[50] on MIPs. Briefly, anisotropy coefficient calculation was performed on a smFISH *z*-stack image, and spot detection on MIPs was performed in 'advanced mode' (no RANSAC, compute min/max intensity from image, use anisotropy coefficient for DoG, add detections to ROI-Manager, mean background subtraction, Sigma = 1.25, DoG and intensity thresholds were manually adjusted). All detected RNA spots were saved as region of interest (ROI) selections and used to create an RNA spots label map image (each spot is identified as a pixel with a distinct value) using a custom Fiji macro. A CellProfiler[51] pipeline was used to segment cells and allocate and count cytoplasmic RNA spots. Briefly, DAPI images were used to identify nuclei as primary objects using a minimum cross-entropy thresholding method, smFISH background fluorescence was used to identify cell boundaries as secondary objects and cytoplasmic regions were derived by subtracting nuclei from cells. The 'RelateObjects' function was used to assign each RNA spot to the mother object cytoplasm. The total number of cytoplasmic RNA spots

per image was computed. To count the number of cytoplasmic RNA spots per cell, cells touching the image border were excluded. The detection of cytoplasmic RNA spots (smFISH) co-localizing with protein spots (IF) was performed manually on chromatic-shift-corrected multichannel *z*-stack images. To avoid operator bias in image annotation, image files were randomized using a custom Fiji macro script before the analysis. The position of cytoplasmic RNA spots was used as reference to check for the presence of resolution-limited particles in the IF channel, distinct from the background and overlapping in *xyz* with the RNA spots. The position of each positive co-localization event was recorded in ROI manager. To account for RNA abundance, the number of RNA spots that co-localized with protein spots was normalized to the total number of cytoplasmic RNA spots per image and expressed as a fraction. If not specified otherwise, images shown in the main figures correspond to representative subsets of single optical planes from chromatic-shift-corrected confocal images. Brightness and contrast adjustments were applied on the entire image in Fiji to facilitate the visualization, without background clipping.

## siRNA transfection

Control (siCTR) and TAF1 siRNAs were purchased from Horizon (ON-TARGETplus Non-targeting Control Pool D-001810-10-05, ON-TARGETplus Human TAF1 siRNA SMARTpool L-005041-00-0010) and resuspended in nuclease-free H₂O. For large-scale transfections, 2.5 × 10⁶ HeLa cells were seeded in 10-cm plates. The next day, cells were transfected using Lipofectamine 2000 (Invitrogen, 11668019) using a low-volume transfection protocol. In brief, after medium removal, cells were treated with 17.5 µL lipofectamine 2000 diluted in 2.8 ml Opti-MEM (Gibco, 31985062) for 15 min at 37 °C. Then, 56 pmol of siRNA diluted in 0.7 ml Opti-MEM was added dropwise to the cells and gently mixed, achieving a 16 nM final siRNA concentration. After ~5 h of incubation at 37 °C, the transfection mix was replaced with prewarmed complete DMEM. Cells were collected 48 h post-transfection.

## Western blot

Samples were loaded on SDS–PAGE gels with 0.5% 2,2,2-trichloroethanol (TCE, Sigma-Aldrich) added for stain-free protein detection[52]. The gel was activated for one minute with UV and the proteins were transferred to a nitrocellulose membrane following standard procedures. Specific proteins were probed with the primary antibodies listed in Supplementary Table 3 and HRP-conjugated secondary antibodies. To reprobe the membrane with an antibody raised in a different species, the previous secondary antibody was inactivated with 10% acetic acid according to[53]. Detection was performed using a ChemiDoc Touch system (BioRad) and images were visualized in ImageLab v6.0 software (BioRad).

## Subcellular fractionation

Adherent cells were washed with cold PBS twice and harvested by scraping on ice. Cell suspension was centrifuged at 400 × g for 5 min at 4 °C and the pellet was resuspended in 4 packed cell volumes (PCV) of hypotonic buffer (50 mM Tris-HCl pH 8.0, 1 mM EDTA, 1 mM DTT, 1× PIC). After 30 min incubation on ice, cells were lysed with 10 hits of Dounce homogenizer and centrifuged at 2,300*g* for 10 min at 4 °C. The supernatant was saved as cytoplasmic extract. Nuclei were washed once in hypotonic buffer and resuspended in 3.5 PCV hypertonic buffer (50 mM Tris-HCl pH 8.0, 0.5 mM EDTA, 500 mM NaCl, 25% glycerol, 1 mM DTT, 1× PIC). Nuclei were lysed with 20 hits of Dounce homogenizer, incubated with agitation for 30 min at 4 °C and centrifuged at 19,000*g* for 30 min at 4 °C. The supernatant was saved as nuclear extract. Cytoplasmic and nuclear extracts were dialyzed against 25 mM Tris-HCl pH 8.0, 5 mM MgCl₂, 100 mM KCl, 10% glycerol, 0.5 mM DTT, 1× PIC at 4 °C using DiaEasy dialyzers (BioVision K1013-10), and protein concentration was measured using the Bradford assay (BioRad, 5000006).

## Immunoprecipitation coupled to LC–MS/MS analysis

Specific and mock (anti-GST) antibodies were coupled either with 200 μL Protein G Sepharose (large-scale IPs, Fig. 4 and Extended Data Fig. 4) or with 2.7 mg Protein G Dynabeads (medium scale IPs, Fig. 6) in IP100 buffer (25 mM Tris-HCl pH 8.0, 5 mM MgCl$_2$, 100 mM KCl, 10% glycerol, 0.1% NP-40, 0.5 mM DTT, 1× PIC) in agitation for 1 h at RT. Antibody-coupled beads were washed twice in IP500 buffer (25 mM Tris-HCl pH 8.0, 500 mM KCl, 5 mM MgCl$_2$, 10% glycerol, 0.1% NP-40, 0.5 mM DTT, 1× PIC) and three times in buffer IP100. Antibody-coupled beads were incubated with cytoplasmic (3–30 mg, medium–large-scale IPs) or nuclear (1–10 mg) extracts overnight at 4 °C. The day after, beads were washed twice with IP500 for 5 min at 4 °C and three times with IP100. Immunopurified proteins were eluted in 0.1 M glycine pH 2.7 and immediately buffered with 1 M Tris-HCl pH 8.0. Eluates were precipitated with TCA (Merck, T0699) overnight at 4 °C and centrifuged at 14,000 $g$ for 30 min at 4 °C. Protein pellets were washed twice with cold acetone and centrifuged at 14,000 $g$ for 10 min at 4 °C. Pellets were denatured with 8 M urea (Merck, U0631) in 0.1 M Tris-HCl, reduced with 5 mM TCEP for 30 min and alkylated with 10 mM iodoacetamide (Merck, I1149) for 30 min, with protection against light. Both reduction and alkylation were performed at RT and in agitation. Double digestion was performed with endoproteinase Lys-C (Wako, 125-05061) at a 1:100 ratio (enzyme:protein) in 8 M urea for 4 h, followed by an overnight modified trypsin digestion (Promega, V5113) at a 1:100 ratio in 2 M urea for 12 h.

Samples were analyzed using an Ultimate 3000 nano-RSLC coupled in line, via a nano-electrospray ionization source, with the LTQ-Orbitrap ELITE mass spectrometer (Thermo Fisher Scientific) or with the Orbitrap Exploris 480 mass-spectrometer (Thermo Fisher Scientific) equipped with a FAIMS (high Field Asymmetric Ion Mobility Spectrometry) module. Peptide mixtures were injected in 0.1% TFA on a C18 Acclaim PepMap100 trap-column (75 μm ID × 2 cm, 3 μm, 100 Å, Thermo Fisher Scientific) for 3 min at 5 μL min$^{-1}$ with 2% ACN and 0.1% FA in H$_2$O and then separated on a C18 Acclaim PepMap100 nano-column (75 μm ID × 50 cm, 2.6 μm, 150 Å, Thermo Fisher Scientific) at 300 nL min$^{-1}$, at 40 °C with a 90 min linear gradient from 5% to 30% buffer B (A: 0.1% FA in H$_2$O/B: 80% ACN, 0.1% FA in H$_2$O), with regeneration at 5% B. Spray voltage was set to 2.1 kV, and the heated capillary temperature was set to 280 °C. For Orbitrap Elite, the mass spectrometer was operated in positive ionization mode, in data-dependent mode with survey scans from $m/z$ 350–1,500 acquired in the Orbitrap at a resolution of 120,000 at $m/z$ 400. The 20 most intense peaks from survey scans were selected for further fragmentation in the Linear Ion Trap with an isolation window of 2.0 Da and were fragmented by CID with normalized collision energy of 35% (TOP20CID method). Unassigned and single charged states were excluded from fragmentation. The ion target value for the survey scans (in the Orbitrap) and the MS2 mode (in the linear ion trap) were set to 1E6 and 5E3, respectively, and the maximum injection time was set to 100 ms for both scan modes. Dynamic exclusion was set to 20 s after one repeat count, with mass width at ± 10 ppm. For Orbitrap Exploris 480 MS associated with the FAIMS module, a combination of two compensation voltages, −40 V and −55 V, was chosen, with a cycle time of 1 s for each. For the full MS1 in DDA mode, the resolution was set to 60,000 at $m/z$ 200 and with a mass range set to 350–1400. The full MS AGC target was 300%, with an IT set to Auto mode. For the fragment spectra in MS2, the AGC target value was 100% (Standard) with a resolution of 30,000 and the maximum Injection Time set to Auto mode. Intensity threshold was set at 1E4. Isolation width was set at 2 $m/z$ and normalized collision energy was set at 30%. All spectra were acquired in centroid mode using positive polarity. Default settings were used for FAIMS with voltages applied as described previously, and with a total carrier gas flow set to 4.2 L min$^{-1}$.

## Mass spectrometry data analysis

Proteins were identified by database searching using SequestHT (Thermo Fisher Scientific) with Proteome Discoverer 2.4 software (PD2.4, Thermo Fisher Scientific) on human FASTA database downloaded from UniProt (reviewed, release 2021_06_03, 20380 entries, https://www.uniprot.org/). Precursor and fragment mass tolerances were set at 7 ppm and 0.6 Da, respectively, and up to 2 missed cleavages were allowed. For the data acquired on the Orbitrap Exploris 480, the software Proteome Discoverer 2.5 version was used with a human fasta database from UniProt (reviewed, release 2022_02_21, 20291 entries). Precursor and fragment mass tolerances were set at 10 ppm and 0.02 Da respectively, and up to 2 missed cleavages were allowed. For all the data, oxidation (M, +15.995 Da) was set as a variable modification, and carbamidomethylation (C, + 57.021 Da) as a fixed modification. Peptides and proteins were filtered with a false discovery rate (FDR) at 1%. Label-free quantification was based on the extracted ion chromatography intensity of the peptides. All samples were measured in technical triplicates. The measured extracted ion chromatogram (XIC) intensities were normalized on the basis of median intensities of the entire dataset to correct minor loading differences. For statistical tests and enrichment calculations, not detectable intensity values were treated with an imputation method, where the missing values were replaced by random values similar to the 10% of the lowest intensity values present in the entire dataset. Unpaired two-tailed $t$-tests, assuming equal variance, were performed on obtained $\log_2$ XIC intensities. Normalized spectral abundance factors (NSAF) were calculated for each protein, as previously described[54]. To obtain spectral abundance factors (SAF), spectral counts identifying a protein were divided by the protein length. To calculate NSAF values, the SAF values of each protein were divided by the sum of SAF values of all detected proteins in each run. All raw LC–MS/MS data have been deposited to the ProteomeXchange via the PRIDE repository with identifier PXD036358.

## Crosslinking-MS metanalysis, protein sequence analysis and modeling

For the metanalysis on the available crosslinking-MS experiments performed on human TFIID, we retrieved and combined the curated datasets from ref. 18 (one dataset, apo-TFIID), ref. 19 (one dataset, apo-TFIID) and ref. 10 (five datasets of TFIID incorporated in preinitiation complex variants: cPICscp, cPICpuma, mPICscp, hPICscp, p53hPIChdm2), for a total of seven datasets. We included only intra and interprotein crosslinks involving TAF1 and found in at least two different datasets. If a crosslink was present only among the Chen et al.[10] datasets, it was considered only if it scored as significant in more than one dataset (probability score < 0.05). The resulting subset of common TAF1 crosslinks is reported in Supplementary Table 2.

TAF1 conservation and structural disorder prediction were computed using ConSurf[55] and Metapredict[56], respectively. The TAF1 full-length model, corresponding to the UniProt entry P21675, was downloaded from the AlphaFold Protein Structure Database (https://alphafold.ebi.ac.uk/). For visual clarity in Figure 5c, the model backbone was manually extended at low-confidence coil regions in UCSF ChimeraX[57]. The TAF1–TBP–TAF11–TAF13 subcomplex was modeled using AlphaFold2_advanced ColabFold implementation with standard settings (https://github.com/sokrypton/ColabFold/)[58] and using the following protein fragments as input: TAF1 (1–300 aa), TBP (150–339 aa), TAF11 (50–211 aa), TAF13 (full length). The TAF1–TAF6$^{HEAT}$–TAF6$^{HEAT}$–TAF8 subcomplex was modeled using AlphaFold2 Multimer extension on COSMIC2 server with standard settings[45,59] and using the following protein fragments as input: TAF1 (300–550 aa), TAF6 (215–482 aa) and TAF8 (130–220 aa). All structural models were visualized, analyzed and rendered in UCSF ChimeraX[57].

## Reporting summary

Further information on research design is available in the Nature Portfolio Reporting Summary linked to this article.

## Data availability

*Homo sapiens* FASTA database from UniProt (https://www.uniprot.org/, reviewed, releases 2021_06_03 and 2022_02_21) was used as reference database for mass-spectrometry protein identification. LC–MS/MS data have been deposited at PRIDE repository with the identifier PXD036358. The AlphaFold protein structure database (https://alphafold.ebi.ac.uk/) was used to download human TAF1 structural prediction file (accession P21675). The ribosome footprinting data plot was obtained from RiboCrypt browser (https://ribocrypt.org/) using human *TAF1* transcript accession ENST00000373790 selecting 'all_merged-Homo_sapiens' as experiment. This paper does not report original code. Any additional information required to reanalyze the data reported in this paper is available from the lead contact (L.T., laszlo@igbmc.fr) upon request. Source data are provided with this paper.

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

## Acknowledgements

We are grateful to the IGBMC Proteomics and Photonic Microscopy platforms and cell culture facility for their assistance and instrumentation. We thank F. Mueller for his helpful advice on imaging data. We thank D. Devys, S. Vincent and D. Helmlinger for thoroughly reading the manuscript and the Tora lab members for helpful discussions. This work was supported by the Fondation ARC pour la recherche sur le cancer (ARCPOST-DOC2021080004113 fellowship awarded to A.B.). The work of the Tora lab (L.T.) was financially supported by grants from Agence Nationale de la Recherche (ANR) ANR-19-CE11-0003-02, ANR-PRCI-19-CE12-0029-01, ANR-20-CE12-0017-03, NIH MIRA grant (R35GM139564), and NSF (Award Number:1933344) grants. This work, as part of the ITI 2021-2028 program of the University of Strasbourg, was also supported by IdEx Unistra (ANR-10-IDEX-0002), and by SFRI-STRAT'US project (ANR 20-SFRI-0012) and EUR IMCBio (ANR-17-EURE-0023) under the framework of the French Investments for the Future Program. The research of S.A., P.K.M.S. and H.T.M.T. was financially supported by the grants from the Deutsche Forschungsgemeinschaft (DFG, German Research Foundation) with the project-IDs 192904750-SFB 992 and TI688/1-1.

## Author contributions

A.B., H.T.M.T. and L.T. conceived and designed the research. A.B., P.M., I.K., E.S., S.A., P.K.M.S. and G.Y. conducted experiments. A.B., P.M., B.M. and L.T. analyzed and interpreted the results. H.T.M.T. and L.T. supervised the study. A.B. and L.T. wrote the manuscript.

## Competing interests

The authors declare no competing interests.

## Additional information

**Extended data** is available for this paper at https://doi.org/10.1038/s41594-023-01026-3.

**Correspondence and requests for materials** should be addressed to László Tora.

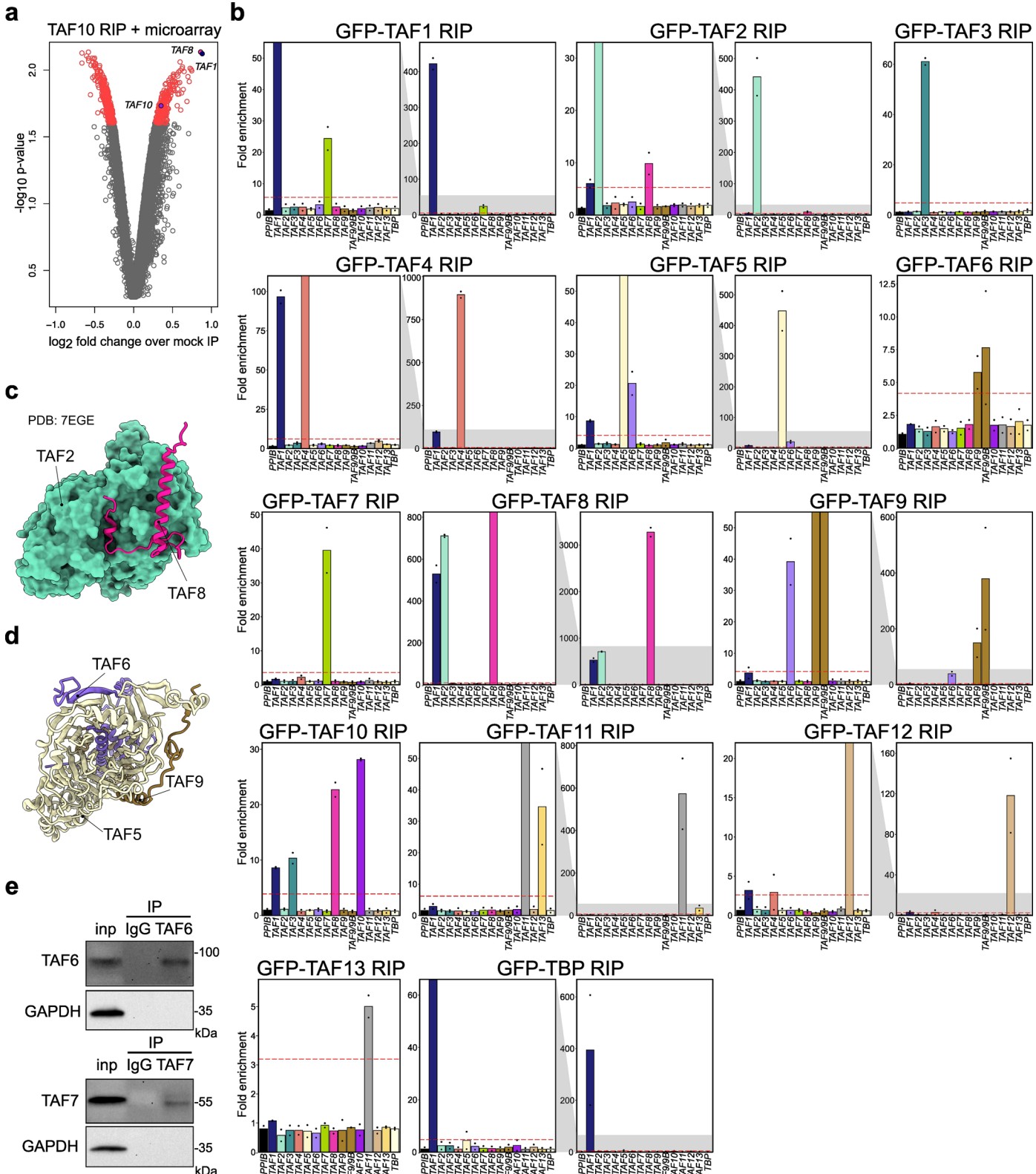

**Extended Data Fig. 1 | RNA immunoprecipitation (RIP) experiments to explore co-translational interactions in TFIID. a**, Volcano plot of endogenous TAF10 RIP-microarray results. *TAF1*, *TAF8* and *TAF10* hits are highlighted. p-values are obtained using the fold change rank ordering statistics method using the fcros R package[24]. **b**, RT-qPCR results of the systematic GFP-RIP assay summarized in Fig. 1e. Each GFP-tagged TFIID subunit was used as bait in a GFP-RIP assay from polysome extracts and enrichment for TFIID subunits mRNAs was assessed by RT-qPCR. Data are expressed as mRNA fold enrichment over mock IP. When necessary, the left panel is the zoomed version of the indicated

grey-shaded area of the full-range plot (right panels). Data points correspond to biological replicas (N=2). The red dashed line threshold corresponds to 4-fold the enrichment level of the negative control target (*PPIB*). **c**, The interaction interface between TAF2/TAF8 as mapped in the TFIID Cryo-EM structure (PDB: 7EGH). The rest of TFIID subunits are not shown for clarity. **d**, Same as in (c) but for TAF5/TAF6. TAF6 completes the β-propeller blade of TAF5 WD40 domain. **e**, Western blot analysis validating the enrichment of the targeted subunit in RIP experiments against endogenous TAF6 and TAF7 from HeLa cells polysome extracts (related to Fig. 1f, g).

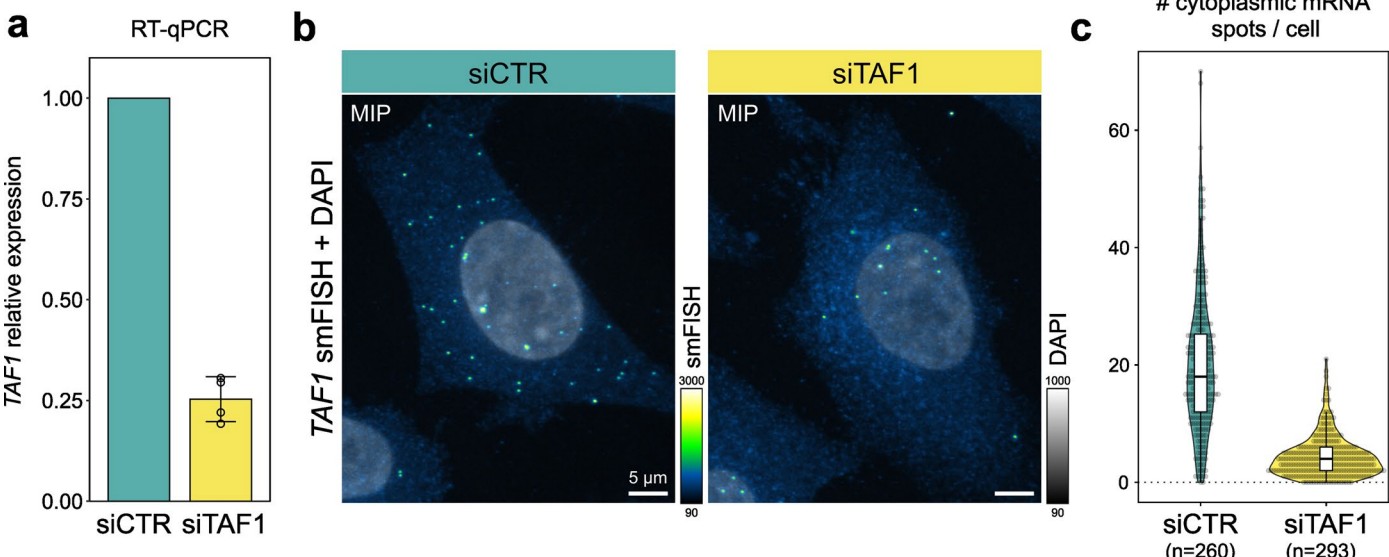

**Extended Data Fig. 2 | *TAF1* smFISH probes validation. a**, *TAF1* siRNA-mediated KD assessed with RT-qPCR and expressed relative to control siRNA (CTR). Data points correspond to four biological replicas. Data presented as mean value +/− SD. **b**, Representative confocal maximum intensity projections (MIPs) of *TAF1* smFISH on HeLa cells transfected with control siRNA (siCTR) or siRNA directed against *TAF1* (siTAF1). smFISH and DAPI channels are displayed using the green fire blue and grayscale color scales, respectively. **c**, *TAF1* KD quantification. Violin plot representing the absolute number of cytoplasmic mRNAs per cell (total number of analyzed cells is in brackets). The boxplot shows the median, 25th and 75th percentile box bounds, and whisker limits as 1.5× interquartile range.

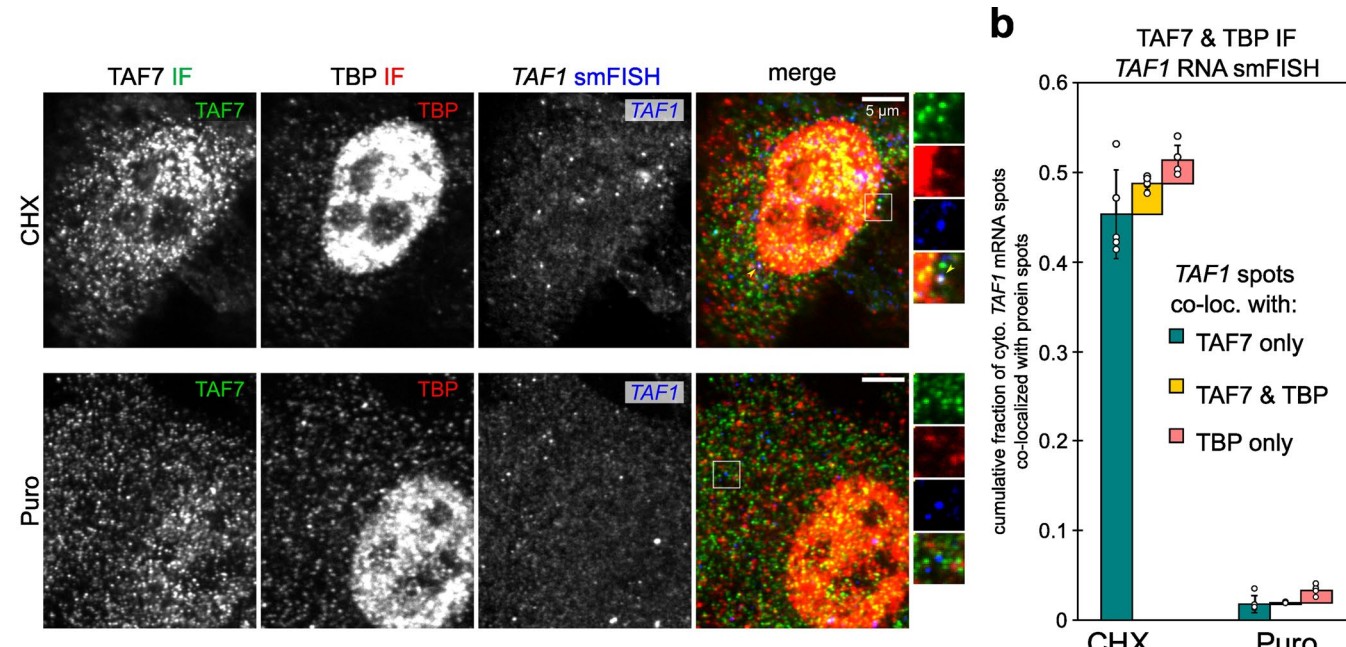

**Extended Data Fig. 3 | *TAF1* RNA smFISH coupled with TBP and TAF7 dual colour immunofluorescence. a**, Representative multicolor confocal images of HeLa cells probed for TAF7 and TBP immunofluorescence (IF) coupled to *TAF1* RNA smFISH. TAF7 protein IF, TBP protein IF and *TAF1* mRNA detection in the merged image are shown in green, red and blue, respectively. Each image is a single confocal optical slice. CHX: cycloheximide; Puro: puromycin.

Triple co-localized spots are indicated by yellow arrowheads. Zoom-in regions (white squares) are shown on the right. **b**, Quantification of the cumulative fraction of *TAF1* mRNAs co-localized with protein signals (TAF7, TBP or both) for each experimental condition. Bars and error bars correspond to mean and SD, respectively (N=5 for CHX; N=4 for Puro; where N corresponds to an independent field of view; total number of cells analysed is in brackets).

## a

### IP-MS on nuclear extracts

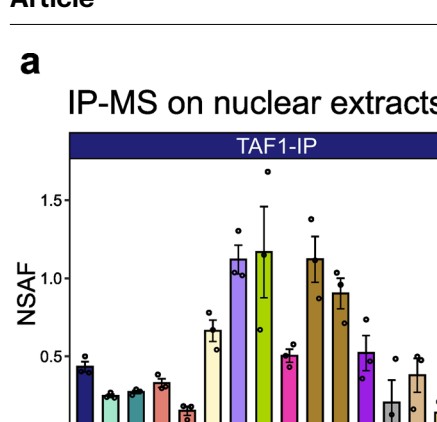
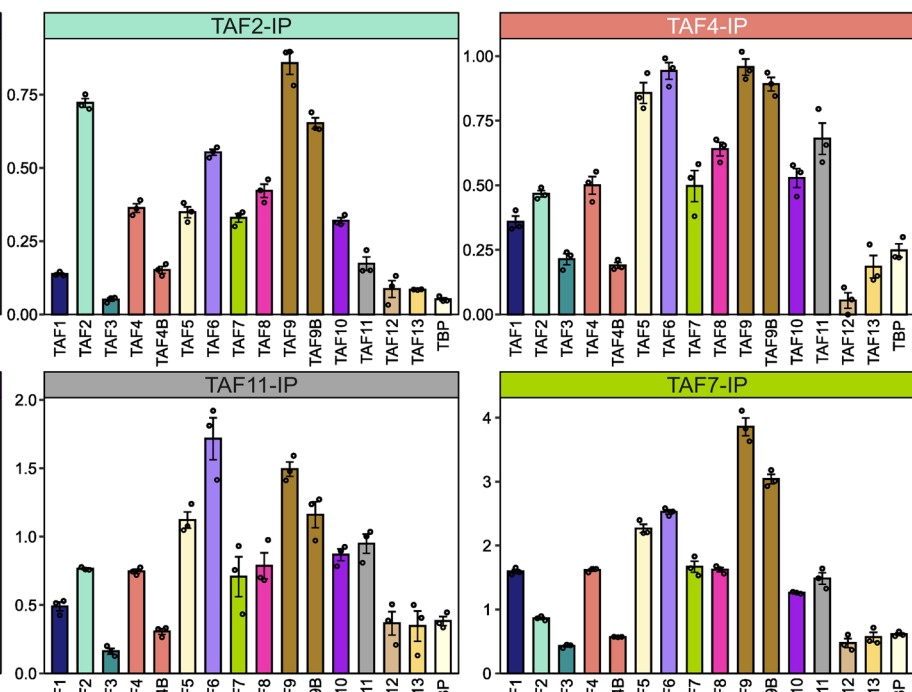

## b cytoplasmic TAF7 IP-MS

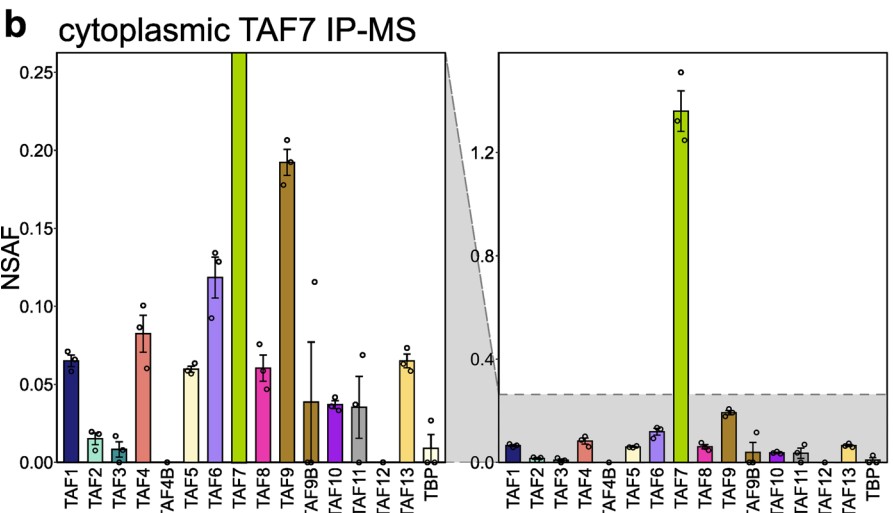

**Extended Data Fig. 4 | Endogenous TFIID subunits immunoprecipitation coupled to mass spectrometry. a**, Immunoprecipitation (IP) of endogenous TFIID subunits coupled to label-free mass-spectrometry performed on human HeLa cells nuclear extracts. Bar plots represent the average NSAF (normalized spectral abundance factor) value for each detected subunit in technical

triplicates. Error bars represent SEM. **b**, Zoomed version of cytoplasmic TAF7-IP bar plot shown in Fig. 4e to better appreciate the retrieved TFIID subunits. Bars represent the average NSAF value for each detected subunit in technical triplicates. Error bars represent SEM.

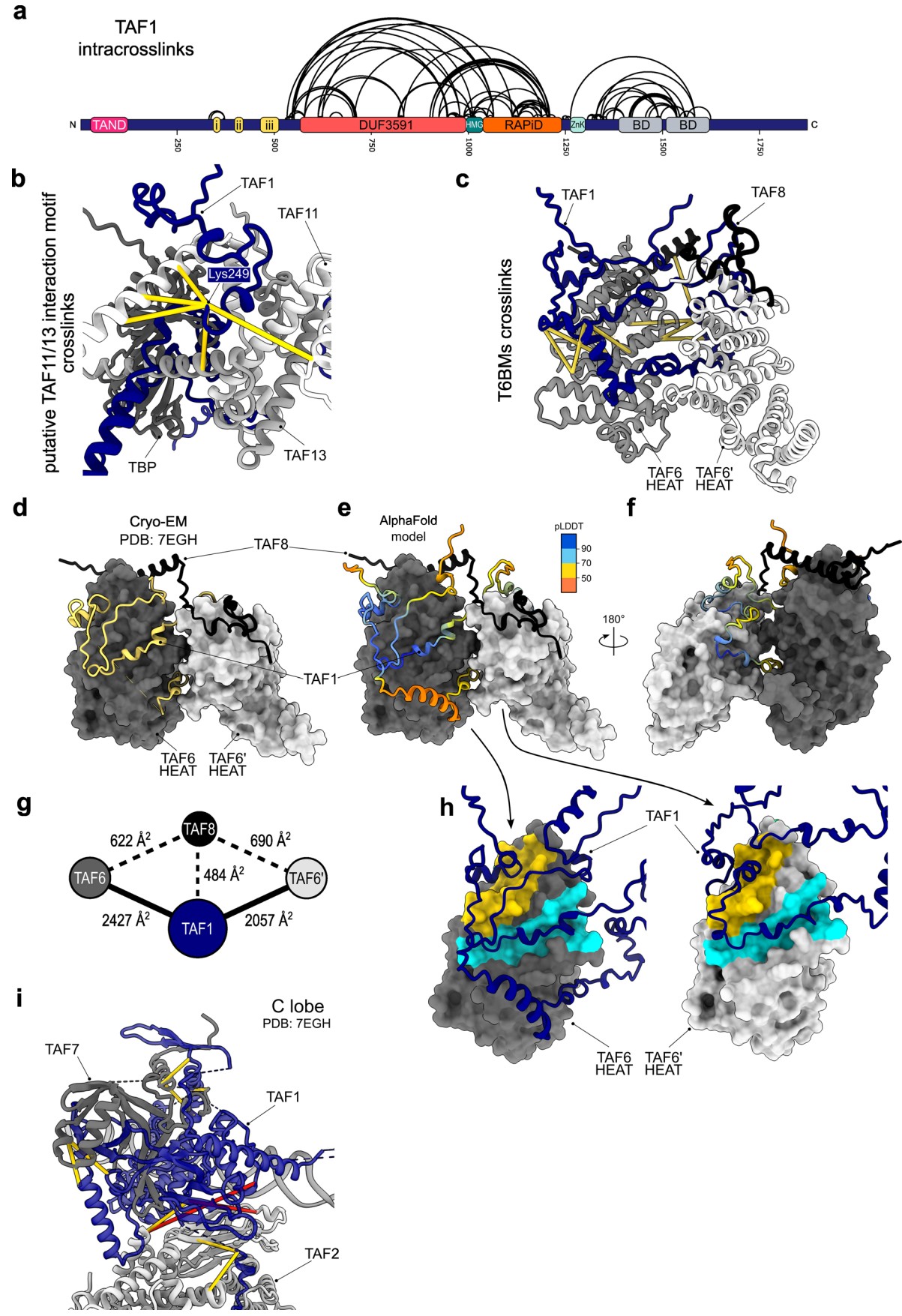

**Extended Data Fig. 5 | See next page for caption.**

**Extended Data Fig. 5 | Structural insights on TAF1 interaction hotspots.**
**a**, Summary of TAF1 intracrosslinks from crosslinking-mass spectrometry metanalysis derived from three independent studies[10,18,19]. Only crosslinks reported in at least two independent datasets are shown. **b**., Interprotein crosslinks of TAF1 Lys249 are mapped on the AlphaFold (AF) model of the TAF1$_{1-300}$/TAF11$_{50-211}$/TAF13/TBP$_{150-339}$ complex. **c**, Mapping of interprotein crosslinks between TAF1 and TAF6/TAF8 in the AF model of TAF1$_{300-550}$/TAF6$^{HEAT}$/ TAF6$^{HEAT}$/TAF8$_{128-218}$ subcomplex. **d**, Cryo-EM structure of TAF1 T6BMs in complex with TAF6 HEAT domains and TAF8 (PDB: 7EGH). The rest of C lobe was removed for clarity. **e**, The AF model described in (c) is shown with TAF1 colored according to pLDDT confidence score and in the same orientation of the experimental structure shown in (d). **f**, 180 degrees rotation of the model shown in (e). **g**, Interface map of the model shown in (e). The size of each node is proportional

to the protein surface area. The values correspond to the buried solvent-accessible surface area between the two connected nodes. Only interfaces with a buried surface area >300 Å$^2$ are shown. TAF1 bridges the two TAF6 HEAT domain copies in the complex. **h**, Equivalent surface patches contacted by TAF1 T6BMs on each of the two copies of TAF6 HEAT domains are highlighted with the same color. In each view the second HEAT domain copy is not shown for clarity. Distinct portions of TAF1 bind to equivalent surfaces on the two copies of TAF6. The representation is based on the model shown in (e). **i**, Interprotein crosslinks of TAF1 are mapped on TFIID C lobe Cryo-EM structure (PDB: 7EGH). For all panels, crosslinks compatible with crosslinker length (Cα-Cα distance < 26 Å) are displayed as yellow pseudobonds. Red pseudobonds correspond to crosslinks that exceed that distance.

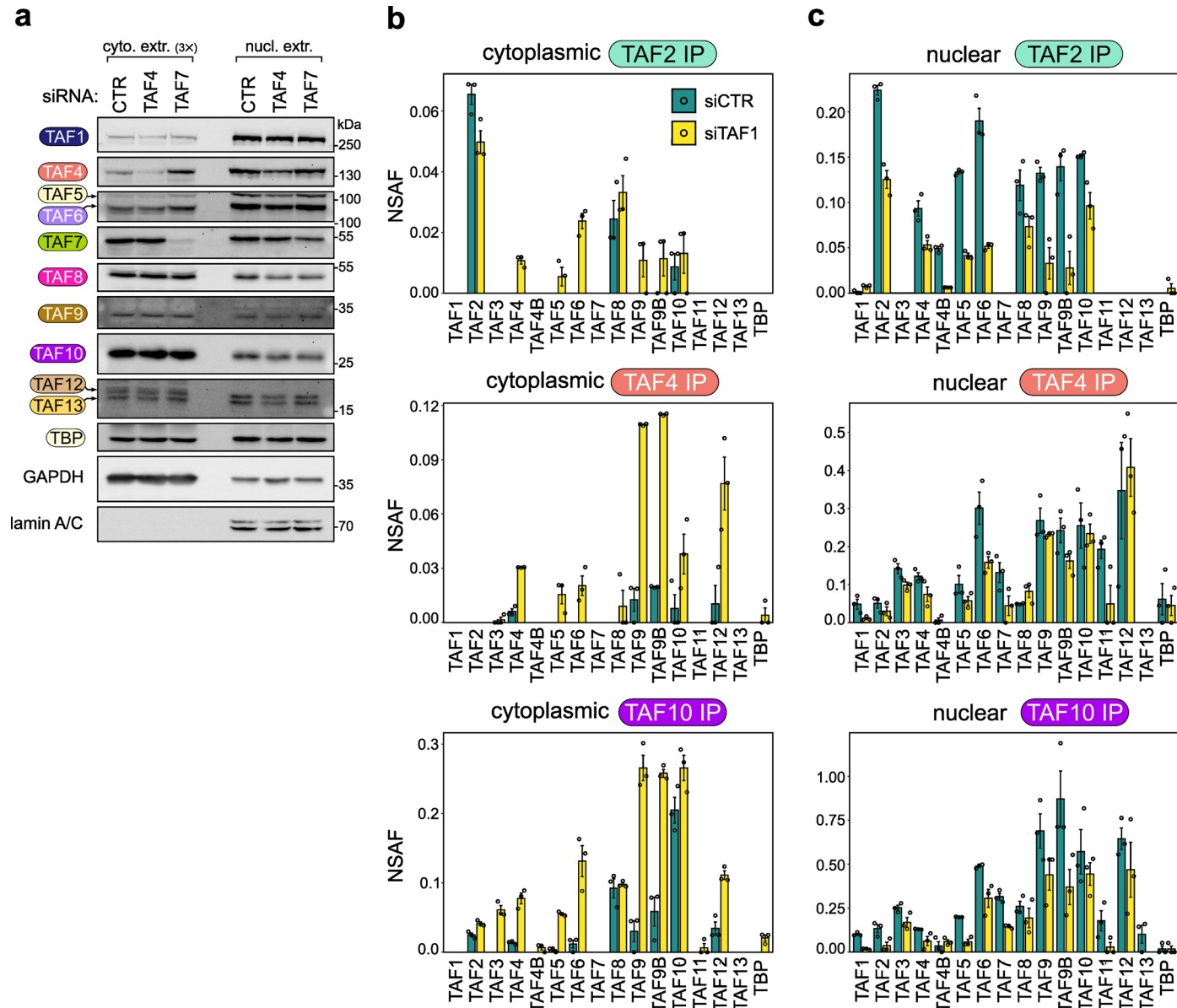

**Extended Data Fig. 6 | Subcellular fractionation upon TAFs knockdown.**
**a**, Subcellular fractionation of HeLa cells transfected with siRNA against *TAF4* or *TAF7* followed by western-blot analysis of endogenous TFIID subunits distribution. GAPDH and lamin A/C were used as loading controls. The amount of loaded cytoplasmic extract is three-times the amount of the nuclear counterpart. **b**, Immunoprecipitation (IP) of endogenous TFIID subunits (TAF2, TAF4, TAF10) coupled to label-free mass-spectrometry (MS) performed on cytoplasmic extracts of HeLa cells upon *TAF1* KD. Bar plots represent the average normalized spectral abundance factor (NSAF) value for each detected subunit in technical triplicates. Error bars represent SEM. **c**, Same as in (b) but the IPs were performed on nuclear extracts. CTR: non-targeting control siRNA.

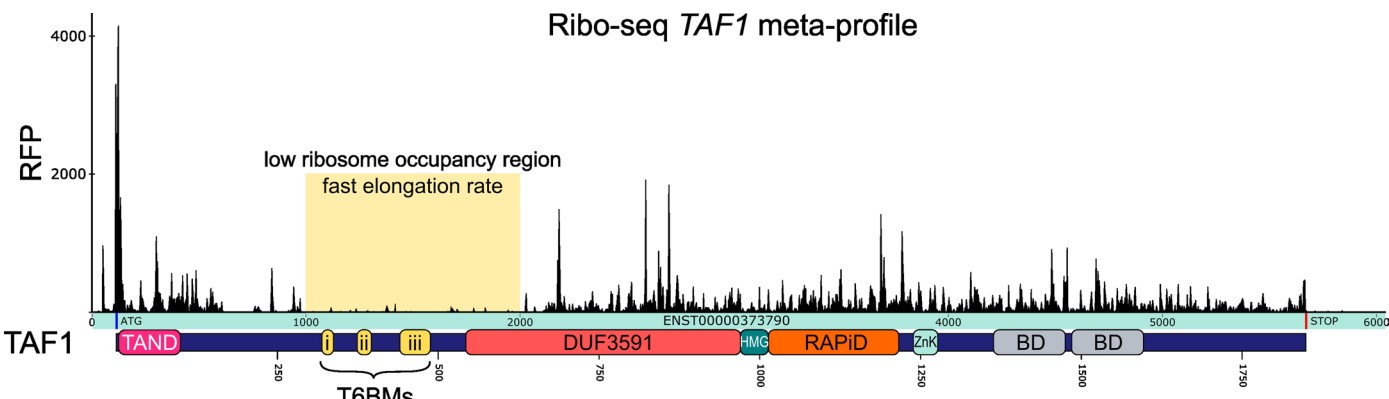

**Extended Data Fig. 7 | *TAF1* ribosome footprinting metaplot.** Ribosome occupancy meta-profile of human *TAF1* derived from merging the available Ribo-seq datasets present in RiboCrypt browser (https://ribocrypt.org/). The yellow window highlights a region of low ribosome occupancy encompassing the three TAF6-binding motifs (T6BMs). Footprints signals coming from reading frames 2 and 3 are omitted for clarity. Below, the TAF1 functional domains are aligned to the CDS. Protein numbering matches the transcript used for this analysis (ENST00000373790). TAF1 domains are shown as in Fig. 5a. RFP, ribosome footprints.

# Reporting Summary

## Statistics

For all statistical analyses, confirm that the following items are present in the figure legend, table legend, main text, or Methods section.

| n/a | Confirmed | |
|---|---|---|
| ☐ | ☒ | The exact sample size (*n*) for each experimental group/condition, given as a discrete number and unit of measurement |
| ☐ | ☒ | A statement on whether measurements were taken from distinct samples or whether the same sample was measured repeatedly |
| ☐ | ☒ | The statistical test(s) used AND whether they are one- or two-sided *Only common tests should be described solely by name; describe more complex techniques in the Methods section.* |
| ☒ | ☐ | A description of all covariates tested |
| ☒ | ☐ | A description of any assumptions or corrections, such as tests of normality and adjustment for multiple comparisons |
| ☐ | ☒ | A full description of the statistical parameters including central tendency (e.g. means) or other basic estimates (e.g. regression coefficient) AND variation (e.g. standard deviation) or associated estimates of uncertainty (e.g. confidence intervals) |
| ☐ | ☒ | For null hypothesis testing, the test statistic (e.g. *F*, *t*, *r*) with confidence intervals, effect sizes, degrees of freedom and *P* value noted *Give P values as exact values whenever suitable.* |
| ☒ | ☐ | For Bayesian analysis, information on the choice of priors and Markov chain Monte Carlo settings |
| ☒ | ☐ | For hierarchical and complex designs, identification of the appropriate level for tests and full reporting of outcomes |
| ☒ | ☐ | Estimates of effect sizes (e.g. Cohen's *d*, Pearson's *r*), indicating how they were calculated |

*Our web collection on statistics for biologists contains articles on many of the points above.*

## Software and code

Policy information about availability of computer code

| | |
|---|---|
| Data collection | RT-qPCR data acquisition was performed using LightCycler® 480 Software (Roche, version 1.5.0). Microscopy data collection was performed using MetaMorph® software (Molecular Devices). Western blot images were acquired using a ChemiDoc Touch system (BioRad). |
| Data analysis | Microscopy data processing and analysis was performed in Fiji (ImageJ version 2.9.0) and CellProfiler (version 4.2.1). Chromatic shift correction was performed using Chromagnon (version v0.90). Western blot images were processed in ImageLab software (BioRad, version 6.0). Mass spectrometry protein identification and analysis was performed using SequestHT in the Proteome Discoverer software (Thermo Fischer Scientific, versions 2.4-2.5). Structural analysis and visualisation was performed in UCSF ChimeraX software (version 1.4). Data rearrangement and plotting was performed in RStudio (R version 4.2.1) with tidyverse package (version 1.3.2). Figures were prepared using Inkscape (version 1.2). This work did not generate any original code relevant for data analysis. |

For manuscripts utilizing custom algorithms or software that are central to the research but not yet described in published literature, software must be made available to editors and reviewers. We strongly encourage code deposition in a community repository (e.g. GitHub). See the Nature Portfolio guidelines for submitting code & software for further information.

## Data

AlphaFold protein structure database (https://alphafold.ebi.ac.uk/) was used to download human TAF1 structural prediction file (accession P21675). Human FASTA database downloaded from UniProt (reviewed, releases 2021_06_03 and 2022_02_21, https://www.uniprot.org/) was used as reference database for mass-spectrometry protein identification. LC-MS/MS data have been deposited at PRIDE repository with the identifier PXD036358 and are publicly available as of the date of publication. Ribosome footprinting data plot was obtained from RiboCrypt browser (https://ribocrypt.org/) using human TAF1 transcript accession ENST00000373790 selecting 'all_merged-Homo_sapiens' as experiment. Any additional information required to reanalyze the data reported in this paper is available from the lead contact (László Tora, laszlo@igbmc.fr) upon request. This paper contains Source Data files.

## Human research participants

| | |
|---|---|
| Reporting on sex and gender | Not relevant to the study. |
| Population characteristics | Not relevant to the study. |
| Recruitment | Not relevant to the study. |
| Ethics oversight | Not relevant to the study. |

Note that full information on the approval of the study protocol must also be provided in the manuscript.

# Field-specific reporting

Please select the one below that is the best fit for your research. If you are not sure, read the appropriate sections before making your selection.

☒ Life sciences   ☐ Behavioural & social sciences   ☐ Ecological, evolutionary & environmental sciences

For a reference copy of the document with all sections, see nature.com/documents/nr-reporting-summary-flat.pdf

# Life sciences study design

All studies must disclose on these points even when the disclosure is negative.

| | |
|---|---|
| Sample size | We planned our experiments in analogy to previously published articles (such as Kamenova et al. 2018). Conclusions drawn in the study were derived by several alternative experimental approaches. No sample-size calculation was performed. |
| Data exclusions | No data were excluded from the analyses. |
| Replication | RIP-qPCR data reported in all the figures were calculated from 2 or 3 biological replicates (specified in figure legends). Western blot of TAF6 and TAF7 antibodies validation for RIP assays shown in Extended Data Figure 1e was performed once. Cell-fractionation upon TAF1 knockdown followed by protein repartition analysis by western blot shown in Figure 6a was performed two times and the results confirmed by mass spectrometry on immunopurified endogenous TFIID subunits shown in Figure 6b. The same experiment upon TAF4 and TAF7 knockdown in Extended Data Figure 6a was performed once. The protocols/experiments provided in the manuscript were reproducible. |
| Randomization | Randomization was not required for the biochemical/molecular experiments. Images for microscopy analyses (immunofluorescence + smFISH) were randomized before manual investigation (blinding). Other analyses were less prone to direct human-based biases thanks to instrument readouts, therefore randomization was not performed. |
| Blinding | The investigator was blinded to samples identity during microscopy data analysis concerning the manual detection of co-localized signals to limit human biases. Other experiments were less prone to direct human-based biases thanks to instrument readouts, therefore blinding was not practiced. |

# Reporting for specific materials, systems and methods

We require information from authors about some types of materials, experimental systems and methods used in many studies. Here, indicate whether each material, system or method listed is relevant to your study. If you are not sure if a list item applies to your research, read the appropriate section before selecting a response.

## Materials & experimental systems

| n/a | Involved in the study |
|---|---|
| ☐ | ☒ Antibodies |
| ☐ | ☒ Eukaryotic cell lines |
| ☒ | ☐ Palaeontology and archaeology |
| ☒ | ☐ Animals and other organisms |
| ☒ | ☐ Clinical data |
| ☒ | ☐ Dual use research of concern |

## Methods

| n/a | Involved in the study |
|---|---|
| ☒ | ☐ ChIP-seq |
| ☒ | ☐ Flow cytometry |
| ☒ | ☐ MRI-based neuroimaging |

## Antibodies

| Antibodies used | See also Supplementary File 1 for a full antibodies table information (with dilution). TAF1 (Abcam, rabbit pAb, ab188427); TAF1 (Abcam, rabbit pAb, ab264327); TAF2 (IGBMC, rabbit pAb, #3038); TAF4 (IGBMC, mouse mAb, 32TA 2B9); TAF5 (IGBMC, mouse mAb, 1TA 1C2); TAF6 (IGBMC, mouse mAb, 25TA 2G7); TAF6 (Bethyl, rabbit pAb, A301-275A); TAF7 (IGBMC, rabbit pAb, #3475); TAF7 (IGBMC, mouse mAb); TAF7 (IGBMC, mouse mAb, 19TA 2C7); TAF8 (IGBMC, rabbit pAb, #3478); TAF9 (Santa Cruz Biotechnology, goat pAb, sc-1248); TAF10 (IGBMC, mouse mAb, 23TA 1H8); TAF10 (IGBMC, mouse mAb, 6TA 2B11); TAF11 (IGBMC, mouse mAb, 15TA 2B4); TAF12 (IGBMC, mouse mAb, 22TA 2A1); TAF13 (IGBMC, mouse mAb, 16TA 3C12); TBP (IGBMC, mouse mAb, 3TF1 3G3); SUPT7L (Bethyl, rabbit pAb, A302-803A); GST (Creative Biolabs, mouse mAb, 15TF2 1D10); lamin A/C (Santa Cruz Biotechnology, mouse mAb, sc-7292); GAPDH (Cell Signaling Technology, rabbit mAb, 14C10); histone H3 (Abcam, rabbit pAb, ab1791); AF488 goat anti-mouse IgG (Life Technologies, goat pAb, A11001); AF488 goat anti-rabbit IgG (Life Technologies, goat pAb, A11008); AF(Plus)647 goat anti-mouse IgG (Life Technologies, goat pAb, A32728); HRP goat anti-mouse IgG (Jackson ImmunoResearch, goat pAb, 115-036-071); HRP goat anti-rabbit IgG (Jackson ImmunoResearch, goat pAb, 111-035-144). |
|---|---|
| Validation | See also Supplementary File 1 for a full antibodies table information (with dilution). Antibody validation references are provided along each antibody used in this study in Supplementary File 1. Also note that antibodies used in immunoprecipitation coupled to mass-spectrometry successfully enriched the target protein. TAF1 (Abcam, ab188427): IF (https://www.abcam.com/products/primary-antibodies/taf1-antibody-ab188427.html) TAF1 (Abcam, ab264327): IP, WB (https://www.abcam.com/products/primary-antibodies/taf1-antibody-ab264327.html) TAF2 (IGBMC, #3038): IP (Trowitzsch et al., 2015) TAF4 (IGBMC, 32TA 2B9): IP, WB (Mohan et al., 2003) TAF5 (IGBMC, 1TA 1C2): WB (Dantonel et al., 1997) TAF6 (IGBMC, 25TA 2G7): WB (Dantonel et al., 1997) TAF6 (Bethyl, A301-275A): IP (https://www.thermofisher.com/antibody/product/TAF6-Antibody-Polyclonal/A301-275A) TAF7 (IGBMC, #3475): IP (Bardot et al., 2017) TAF7 (IGBMC, 31TA 2C12): IP (present work) TAF7 (IGBMC, 19TA 2C7): WB, IP (Lavigne et al., 1996) TAF8 (IGBMC, #3478): WB (Bardot et al., 2017) TAF9 (Santa Cruz Biotechnology, sc-1248): WB (https://datasheets.scbt.com/sc-1248.pdf) TAF10 (IGBMC, 23TA 1H8): IP (Soutoglou et al., 2005) TAF10 (IGBMC, 6TA 2B11): WB (Wieczorek et al., 1998), IF (Soutoglou et al., 2005) TAF11 (IGBMC, 15TA 2B4): IP (Gupta et al., 2017) TAF12 (IGBMC, 22TA 2A1): WB (Malecova et al., 2016) TAF13 (IGBMC, 16TA 3C12): WB (Mengus et al., 1995) TBP (IGBMC, 3TF1 3G3): WB (Brou et al., 1993) SUPT7L (Bethyl, A302-803A): WB, IHC (https://fortis-datasheets.s3.us-east-2.amazonaws.com/A302-803A-1.pdf) GST (Creative Biolabs, 15TF2 1D10): IP (https://www.antibody-creativebiolabs.com/anti-hpgds-antibody-cblg1-2281-87009.htm) lamin A/C (Santa Cruz Biotechnology, sc-7292): WB (https://www.scbt.com/p/lamin-a-c-antibody-636) GAPDH (Cell Signaling Technology, 14C10): WB, IF, IHC (https://www.cellsignal.com/products/primary-antibodies/gapdh-14c10-rabbit-mab/2118) histone H3 (Abcam, ab1791): WB, IP, ChIP, IF (https://www.abcam.com/products/primary-antibodies/histone-h3-antibody-nuclear-marker-and-chip-grade-ab1791.html) |

## Eukaryotic cell lines

Policy information about cell lines and Sex and Gender in Research

| Cell line source(s) | HeLa cells were obtained from IGBMC cell culture facility. HeLa Flp-In/T-REx cells for the generation of GFP-TAF fusion expressing cell lines were not from a commercial source and they were described in van Nuland et al., 2013 (MCB, doi:10.1128/MCB.01742-12) and Antonova et al., 2018 (NSMB, doi:10.1038/s41594-018-0156-z). The resulting engineered GFP-fusion cell lines were described in Antonova, 2020 (doi:10.33540/207). The E14 mouse embryonic stem cells (ES Parental cell line E14Tg2a.4) were obtained from Mutant Mouse Resource and Research Center (MMRRC) (Citation ID:RRID:MMRRC_015890-UCD). |
|---|---|
| Authentication | HeLa cells were authenticated according to ATCC STR profiling. mESCs morphology was periodically controlled. |

| Mycoplasma contamination | All cell lines used in the study were tested for mycoplasma contamination by the IGBMC Cell Culture Facility and were negative. |
| Commonly misidentified lines<br>(See ICLAC register) | No commonly misidentified cell lines were employed for the study. |

