## [Peer Review File · Nature Structural & Molecular Biology]

Peer Review Information

Manuscript Title: Hierarchical TAF1-dependent co-translational assembly of the basal transcription factor TFIID

Corresponding author name(s): Laszlo Tora

Reviewer Comments & Decisions:

Decision Letter, initial version:
--

Message: 5th Jan 2023

Dear Dr. Tora,

Thank you again for submitting your manuscript "Hierarchical TAF1-dependent co-translational assembly of the basal transcription factor TFIID". I apologize for the delay in responding, which resulted from the difficulty in obtaining suitable referee reports. Nevertheless, we now have comments (below) from the 3 reviewers who evaluated your paper. In light of those reports, we remain interested in your study and would like to see your response to the comments of the referees, in the form of a revised manuscript.

You will see that while all 3 reviewers highly appreciate the work, Reviewer #1 requests additional colocalization experiments to conclusively distinguish between possible assembly models. Please be sure to address/respond to all concerns of the referees in full in a point-by-point response and highlight all changes in the revised manuscript text file. If you have comments that are intended for editors only, please include those in a separate cover letter.

We expect to see your revised manuscript within 6 weeks. If you cannot send it within this time, please contact us to discuss an extension; we would still consider your revision, provided that no similar work has been accepted for publication at NSMB or published elsewhere.

Reporting Summary:

Please note that all key data shown in the main figures as cropped gels or blots should be presented in uncropped form, with molecular weight markers. These data can be aggregated into a single supplementary figure item. While these data can be displayed in a relatively informal style, they must refer back to the relevant figures. These data should be submitted with the final revision, as source data, prior to acceptance, but you may want to start putting it together at this point.

Data availability: this journal strongly supports public availability of data. All data used in accepted papers should be available via a public data repository, or alternatively, as Supplementary Information. If data can only be shared on request, please explain why in your Data Availability Statement, and also in the correspondence with your editor. Please note that for some data types, deposition in a public repository is mandatory - more information on our data deposition policies and available repositories can be found below: <https://www.nature.com/nature-research/editorial-policies/reporting->

standards#availability-of-data

We require deposition of coordinates (and, in the case of crystal structures, structure factors) into the Protein Data Bank with the designation of immediate release upon publication (HPUB). Electron microscopy-derived density maps and coordinate data must be deposited in EMDb and released upon publication. Deposition and immediate release of NMR chemical shift assignments are highly encouraged. Deposition of deep sequencing and microarray data is mandatory, and the datasets must be released prior to or upon publication. To avoid delays in publication, dataset accession numbers must be supplied with the final accepted manuscript and appropriate release dates must be indicated at the galley proof stage.

[Redacted]

Sincerely,
Sara

Sara Osman, Ph.D.
Associate Editor
Nature Structural & Molecular Biology

Referee expertise:

Referee #1: Macromolecular complexes

Referee #2: Gene regulation, mass spectrometry

Referee #3: Transcription initiation

Reviewers' Comments:

Reviewer #1:

Remarks to the Author:

The study by Bernardini et al. investigates the biogenesis of TFIID, a conserved and essential multi-subunit transcription factor. By combining molecular biology, imaging and proteomic approaches with structural data mining, the authors convincingly demonstrate that most TFIID subunits assemble in a co-translational manner, expanding their previous observations (authors' ref. #22) to the scale of the entire complex. Strikingly, they also demonstrate that the TAF1 subunit acts as a centerpiece in which dedicated interaction motifs allow the recruitment of distinct TFIID sub-modules during the course of translation. They finally report that the absence of TAF1 leads to the cytoplasmic accumulation of TFIID sub-complexes, supporting a model in which TAF1-dependent holo-complex assembly occurs in a co-translational, hierarchical manner prior to nuclear import.

Overall, the data presented provide important, novel insights into the biogenesis of large multiprotein assemblies. The figures are presented in a logical manner and the conclusions are strongly supported by the wide range of methodologies used by the authors. The manuscript is clearly written and balanced, and accurately references previous knowledge in this emerging field. The attractive model put forward only needs to be reinforced by the following suggested experiments.

1. The authors propose a "sequential N- to C-terminal order of assembly" of TFIID building blocks on the TAF1 polypeptide (Fig. 7 and Discussion). Yet, some subunits (e.g., TBP, TAF4, TAF10), which are recruited by the N-terminal domains of TAF1, colocalize with only a small fraction of TAF1 transcripts (6-10%, Fig. 3). In contrast, a large fraction of TAF1 mRNAs is found to interact with TAF7 (40%, Fig. 3), which binds to the most C-terminal interaction motif in the TAF1 polypeptide (Fig. 5). Could the distinct levels of TAF proteins/TAF1 mRNA colocalization simply reflect different affinities of the corresponding antibodies for their targets in immunofluorescence experiments? In order to compare the extent of co-translational recruitment between the different subunits, could the authors use the same anti-GFP antibodies on different GFP-TAF cell lines? Alternatively, it cannot be excluded that the different TFIID building blocks interact with TAF1 in an independent rather than ordered manner. To discriminate between these two assembly models, it would be interesting to simultaneously probe two distinct TAF subunits for their colocalization with TAF1 mRNAs in microscopy analyses. Could the authors examine whether TBP-containing TAF1 mRNAs foci also contain TAF4, TAF10, TAF7, or other subunits for which compatible antibodies are available?

2. The proteomic analysis of cytoplasmic TFIID entities, in normal conditions (Fig. 4) or upon TAF1 depletion (Fig. 6), strongly supports a model in which the TAF1-dependent assembly step is the last to occur before nuclear import. Consistently, free, full-length TAF1 polypeptides are barely detectable in cytoplasmic extracts (Figs. 4,6). However, TAF1 polypeptides are readily detected by immunofluorescence using antibodies directed against their N-terminus (Fig. 2). Could the authors compare the level of TAF1 detection using N- and C-terminus-directed antibodies in immunofluorescence experiments? This could reinforce the idea that only nascent TAF1 polypeptides are present in the cytoplasm

and that the completion of their synthesis coincides with the assembly and subsequent import of the whole complex.

Minor remarks

- The authors use immunofluorescence to score protein colocalization with mRNAs (Fig. 2-3). It is likely that the corresponding protein spots correspond to several co-localized polypeptides. In this respect, the transcripts that are not detected as co-localized with protein spots could also be translated, yet to an extent that does not allow detection of the corresponding protein (or partners). This should be indicated in the Results section.
- Could the authors comment on the fact that the TAF4/TAF1 mRNA interaction is partially insensitive to puromycin treatment (Fig. 3C)?
- The authors indicate that the TAF1 protein encompasses several predicted ribosome pause motifs (Extended Data Fig. 6). Could they analyze available ribosome footprinting datasets to confirm the existence of such pause sites?
- TAF8>TAF10 (lines 92, 133) should be TAF10>TAF8.
- Fig. S1C (line 164) should be Extended Data Fig. 1C.

Reviewer #2:

Remarks to the Author:

TFIID is a ~1.3 MDa multi-subunit basal transcription factor that initiates the recruitment of RNA polymerase II to the promoter. The complex is composed of TATA box-binding protein (TBP) and 13 different TBP-associated factors (TAFs). Previous observations led to a model that postulates the formation of different TFIID sub-complexes in the cytoplasm, and independent translocation into the nucleus, where the holo-TFIID complex would form. The authors have studied the assembly of endogenous TFIID complexes and sub-complexes in cells, with particular emphasis on the cytoplasm. Using a systematic RNA immunoprecipitation (RIP) experiments in human HeLa FRT cell lines with Dox-inducible N-terminally tagged TAFs, they identified the nascent TAF1 polypeptide as central for TFIID assembly. Using immunofluorescence with antibodies against TAF1, TAF4, TAF7 and TAF10, combined with single-molecule RNA FISH for TAF1 mRNA, they detected co-localization events, in line with the recruitment of TFIID subunits on the nascent TAF1 polypeptide during its synthesis. Immunoprecipitation of TFIID subunits from cytoplasmic extracts, followed by label-free mass spectrometry analysis, resulted in the identification of cytoplasmic multi-subunit TFIID sub-complexes. Analysis of all available crosslinking mass spectrometry experiments using highly purified TFIID or PIC-incorporated TFIID, identified three main cross-linking "hot spot" regions in the TAF1 protein, which may serve as anchoring points for the different pre-assembled TFIID sub-modules. Interestingly, siRNA-mediated TAF1 depletion results in the cytoplasmic accumulation of distinct TAFs. Taken together, the authors propose a multi-step hierarchical model for TFIID biogenesis that concludes with co-translational assembly of the complex on the nascent TAF1 polypeptide.

The study is novel, comprehensive and impressive. The manuscript is well-written and the conclusions drawn are based on multiple lines of evidence. I anticipate it will be of great interest to scientists interested in transcriptional regulation and in co-translation in general. Multiple state-of-the-art complementary methodologies were employed. The quality of the data and presentation is high. Appropriate statistical testing was employed in the mass spectrometry analysis.

Suggestions for future studies:

1. Could Dox-inducible GFP-TAF1 transgenes in which a stop codon will be introduced at different positions along the mRNA be used to check for the existence of intermediate stages of the model proposed in Figure 7?
2. Future studies would benefit from the use of distinct TAFs that are tagged by different fluorescent proteins, and live imaging to determine whether holo-TFIID assembles in the cytoplasm or following nuclear translocation.
3. It would be very interesting to examine whether siRNA-mediated TAF7 knockdown would affect nuclear import of the assembled TFIID.
4. Interestingly, the alpha and beta subunits of the negative cofactor 2 (NC2), also contain histone fold domains (PMID: 8670811). NC2 is a negative regulator of TBP activity that binds TBP-promoter DNA. It would be interesting to explore whether NC2 subunits are co-translated.

Major Comments:

1. It is important to demonstrate the specificity of TAF1 depletion (Figure 6), i.e., does knockdown (KD) of other TAFs result in similar observations?
2. Although the authors mention that each GFP-tagged TAF was previously shown to incorporate into TFIID purified from nuclear extracts, N-terminal tagging of specific TAFs with GFP may be problematic, as some of their functional domains are N-terminally located, e.g., a protein kinase domain in the N-term. of TAF1, which was shown to phosphorylate RAP74 (PMID: 8625415), and histone fold domains in the N-term. of TAF8 and TAF13. Could N-term tagging affect the function or perhaps, the stability of some of the TAF polypeptides, especially the small TFIID subunits?
3. TAF1 depletion leads to accumulation of TFIID subunits in the cytoplasm. It would be interesting to discuss these findings with regards to outcomes of TAF1 KO that were previously reported in the literature.

Minor comments:

1. p. 14 line 376 (Figure 6A) - TAF9 does not appear to be significantly enriched in the cytoplasmic extract following siRNA-mediated TAF1 KD. It is advisable to include a panel with quantitation of the protein levels based on multiple experiments, in addition to representative western blots.
2. The authors should indicate the difference between the inhibitory effects of cycloheximide (which "freezes" translating ribosomes on the mRNA) and puromycin (which releases nascent peptides from ribosomes) on protein synthesis.
3. p.9 line 260 – it would be better if the list of TAF4-interacting TAFs would be ordered based on NSAF values, rather than ascending numerical order.
4. For the benefit of readers who are unfamiliar with the pLDDT scores, the authors should relate to the pLDDT values (e.g. pLDDT > 90 are expected to be modelled to high accuracy, pLDDT between 70 and 90 are expected to be modelled well and pLDDT between 50 and 70 are low confidence and should be treated with caution).

5. p.14 line 368: I suggest rephrasing: "This would allow TFIID assembly on the N-terminal half of TAF1 before the protein is released from the ribosome." to:
This would potentially allow TFIID assembly on the N-terminal half of TAF1 before the protein is released from the ribosome.

6. It would be interesting to discuss the identification of distinct TAF4B- and TAF9B-containing complexes (Figure 4).

7. p.13 lines 338-339 – reference to PMID: 8170939 should be added.

8. Typos:

- p.11, 13, 39, Figure 5: pLDTT should be replaced by pLDDT
- p. 16 line 424 add: it - we find it remarkable...
- p.16 line 425 instead of: This point - This points
- p. 35 Figure legend: D Same as in C (not B)

Reviewer #3:

Remarks to the Author:

RE: NSMB-A46922

The authors previously elegantly demonstrated co-translational assembly of the TAF8-10 TFIID subcomplex as well as some parts of other complexes, namely TREX-2 and SAGA. Here they report a larger scale approach that revealed co-translational assembly to occur all TFIID subcomplexes. Notably, final assembly of canonical TFIID appears to be coupled with TAF1 translation. This step also appears to be the rate limiting step in canonical TFIID assembly and may help to minimize off pathway interactions. Co-translational "gluing" would be a new function of TAF1 in TFIID biology. The paper also reveals in new detail how co-translation may be a way to ensure proper assembly of large multi-protein complexes.

This manuscript proposes an elegant model for canonical TFIID assembly that is supported by multiple independent lines of evidence. The authors argue, and I agree, that the proposed mechanism of co-translational complex assembly may also be relevant for other complexes. Evidence for that is also provided in a prior publication (PMID: 30988355). Unfortunately, I personally lack sufficient knowledge of the literature to evaluate how novel or generalizable this "co-translational" aspect of their finding is or if something similar has been previously reported for another multi protein complexes. The model, as well as the novel role for TAF1, however, should be of broad relevance to the scientific community.

Overall, the data are well presented and conclusions are largely supported by multiple independent lines of experimental evidence. My two major concerns with this manuscript are that 1) the figures would be easier to follow if they would contain a little more description and that 2) it is hard to evaluate the relative importance of measurements.

1) The authors make use of a broad array of methods, which makes the manuscript both elegant and convincing. But it also makes it likely that readers are not familiar with the one or another approach. To help readers staying engaged simplified descriptions of how

the assays work would help. Specifically: A brief description that puromycin terminates translation while cycloheximide stalls it would make Fig.1 more instantly understandable for folks outside the field without the need to read the text or consult google. Fig.2C "white dots indicate an overlap of TAF1 protein and mRNA, either TAF1 or negative control CTNNB1". [Site note: In Fig 3, I find the zoom out way more informative, potentially due to printing quality. Consider zooming in?]. 5B: I am not sure a reader should be expected to know what pLDTT is? Maybe add a sentence why you think this is important and what it does. If space is limiting, maybe the discussion could be shortened a little to accommodate 1-2 more sentences per figure.

2) To improve evaluation of the relative importance of RIPed mRNAs it would be helpful to supplement qPCR with a less biased approach. One option could be to subject the IPed RNA to total RNA-seq (Ribo0). This approach could also reveal the absolute rank of TAF1 mRNA in each IP and if there may be another potentially non-TAF that may have been missed with qPCR. This or any other means to help better evaluate how impactful the measures are would help. I.e. how significant is 3% TAF1 in Fig1 F/G ?

This issue of "relative importance" is also relevant to Fig. 3. TBP, for example, shows a very low colocalization with TAF1 mRNA, despite binding N-terminal and thus potentially "earlier" co-translationally than TAF7. This could be as TBP also functions in other subcomplexes. Yet TAF7 is in 40% and TAF4 10% etc. What conclusions can we draw from this quantification? May it be more informative to plot what % of TAF1 mRNA has the IF protein (i.e. flip it?).

Additional comments:

An exciting, somewhat left open question is whether TFIID co-translational assembly is ordered (controlled), random/chaotic or a mix of both. Some subcomplexes, i.e. TAF6-9,11-13,2-8 appear to be co-translationally reciprocal while most seem directional. The reciprocal nature of some may argue that certain interactions just occur early and by chance and have a low off rate. If co-translational assembly is evolutionary favored, one may expect to find the protein interaction domains enriched in N-terminal regions, rather than C-terminal ones. Furthermore, analogous to colinear expression of HOX gene clusters, if co-translational assembly is crucial, it could be expected that proteins with multiple interaction domains like TAF1 have them in the order of co-translational assembly. Last, I may be wrong here but it appears that there was more success tagging TAFs N-terminal than C-terminal. Can the authors comment how these observations align with their model? Does their domain analysis resonate with this speculation?

Given the authors expertise, I feel it would also be appropriate to ask the authors to discuss how far their finding aligns with (their and others) previously reported non-canonical TFIID complexes (i.e. in stem cells).

Acknowledging that translation goes N to C terminal, given the importance the results have on the authors conclusions, I wonder if it was attempted to use the N terminal GFP-TAF1 or the antibody from 2A to confirm the findings in 4F. Would the authors now detect TBP? 11/13?

Maybe consider drawing the grey ball in 4A as a multi protein complex, i.e. several balls?

The list of primers and used antibodies with # is very useful. The detailed method section is laudable!

Sascha H. Duttke

Author Rebuttal to Initial comments

NSMB-A46922 Response to Referees

Note: Our responses are marked in blue. Page numbering refers to the marked version of the revised manuscript.

Reviewer #1

Remarks to the Author:

Overall, the data presented provide important, novel insights into the biogenesis of large multiprotein assemblies. The figures are presented in a logical manner and the conclusions are strongly supported by the wide range of methodologies used by the authors. The manuscript is clearly written and balanced, and accurately references previous knowledge in this emerging field.

We were very happy to learn that the Reviewer found that our conclusions were strongly supported by the data and that the manuscript was well written and balanced.

The attractive model put forward only needs to be reinforced by the following suggested experiments. **1.** The authors propose a “sequential N- to C-terminal order of assembly” of TFIID building blocks on the TAF1 polypeptide (Fig. 7 and Discussion). Yet, some subunits (e.g., TBP, TAF4, TAF10), which are recruited by the N-terminal domains of TAF1, colocalize with only a small fraction of TAF1 transcripts (6-10%, Fig. 3). In contrast, a large fraction of TAF1 mRNAs is found to interact with TAF7 (40%, Fig. 3), which binds to the most C-terminal interaction motif in the TAF1 polypeptide (Fig. 5).

Could the distinct levels of TAF proteins/TAF1 mRNA colocalization simply reflect different affinities of the corresponding antibodies for their targets in immunofluorescence experiments? In order to compare the extent of co-translational recruitment between the different subunits, could the authors use the same anti-GFP antibodies on different GFP-TAF cell lines?

The reviewer is right by stating that distinct degree of endogenous TAF proteins/TAF1 mRNA colocalization may simply reflect different affinities of the corresponding antibodies for their targets in IF experiments. To overcome the potential difference in antibody detection sensitivity during immunofluorescence, originally, we followed exactly the strategy suggested by the Reviewer using the exogenously expressed GFP-tagged TAF system. Unfortunately, while proven excellent for bulk RIP assays performed on polysome extracts, the inducible GFP-fusion cell lines showed an intrinsic cell-to-cell variability in expression levels. This variable GFP-TAF expression makes the GFP imaging far from being ideal for single-spot detection and counting. Below we provide example images of one such cell line (GFP-TAF8) visualized with anti-GFP immunofluorescence (**Fig. R1**). Upon induction, cells express different amounts of the fusion protein, making the single-cell measurements difficult to normalize. A second, technical challenge is that the cytoplasmic fluorescence pattern observed in this setup in individual cells cannot be resolved (see right-end upper panel in **Fig. R1**) making the single-spot analysis practically impossible with our imaging setup. For these reasons, we directed our attention towards endogenous subunits, with the additional benefit of moving away from an overexpression system.

Figure Rebuttal R1. GFP-TAF8 HeLa FRT-TO cells stained by GFP immunofluorescence (IF) and DAPI for nuclear counterstain. Expression of the transgene is induced by doxycycline (Dox) treatment. The first two panels were imaged in epifluorescence microscopy. The right-end panel was imaged by confocal microscopy (scale bars = 20 μm).

Alternatively, it cannot be excluded that the different TFIID building blocks interact with TAF1 in an independent rather than ordered manner. To discriminate between these two assembly models, it would be interesting to simultaneously probe two distinct TAF subunits for their colocalization with TAF1

mRNAs in microscopy analyses. Could the authors examine whether TBP-containing TAF1 mRNAs foci also contain TAF4, TAF10, TAF7, or other subunits for which compatible antibodies are available?

We thank the Reviewer for her/his excellent suggestion. As suggested, we asked whether TBP-containing *TAF1* mRNA foci would also contain a second TAF. To this end, we simultaneously probed for TBP and TAF7 by immunofluorescence and *TAF1* mRNA smFISH. We could carry out this combination due to the species of origin of the two antibodies (mouse anti-TBP and rabbit anti-TAF7 antibodies, respectively). Moreover, TBP and TAF7 recognize the first and last interaction hotspot along TAF1, representing the two “extremes” in case of sequential assembly. These novel results are now included in a new **Extended Data Fig. 3** of the revised ms and described on page 9 (Main text) of the Results section. We observed puromycin-sensitive double co-localized spots with *TAF1* mRNA, further confirming the existence of multi-protein assembly intermediates on nascent TAF1 polypeptide. As correctly anticipated by the Reviewer, the high levels of spatial co-association of TAF7 with *TAF1* mRNA could hint at an independent association of the different TFIID building blocks on nascent TAF1. Additionally, we also detected a fraction of TBP-associated *TAF1* foci lacking TAF7 signal. Overall, based on these new observations, we highlighted the non-obligate nature of the sequential assembly model in the Discussion section (Main text, page 17). In addition, we tuned down the sequential step-wise assembly model in the Discussion section, without mentioning the word “sequential” in the Title, the Abstract or the Results sections. Along the same lines, we renamed the TAF1 interaction hotspots using an alphabetical (A-B-C) rather than numeric (1-2-3) notation, both in the text and in the figures.

2. The proteomic analysis of cytoplasmic TFIID entities, in normal conditions (Fig. 4) or upon TAF1 depletion (Fig. 6), strongly supports a model in which the TAF1-dependent assembly step is the last to occur before nuclear import. Consistently, free, full-length TAF1 polypeptides are barely detectable in cytoplasmic extracts (Figs. 4,6). However, TAF1 polypeptides are readily detected by immunofluorescence using antibodies directed against their N-terminus (Fig. 2). Could the authors compare the level of TAF1 detection using N- and C-terminus-directed antibodies in immunofluorescence experiments? This could reinforce the idea that only nascent TAF1 polypeptides are present in the cytoplasm and that the completion of their synthesis coincides with the assembly and subsequent import of the whole complex.

We thank the Reviewer for highlighting this possibility. Note that a weak full length TAF1 signal is readily detected in cytoplasmic extracts also by western blot (Fig. 6a) and a low amount of TAF1 is also detected in cytoplasmic extract in our TAF7 IP (**Extended Data Fig. 4b**). In addition, we acknowledged potential technical limitations of our IP-MS experiments stating that “*the abundance of TAF1 mature protein in the cytoplasm is below the detection limit in this analysis*” (Main text, page 11; see also our answer to Reviewer 3, Additional Comment 3). As suggested by the Reviewer, we performed the direct comparison

of N- vs C-ter-directed TAF1 antibodies staining by immunofluorescence. We quantified the fluorescence in the cytoplasmic vs nuclear volume in confocal Z-stacks. The results of this analysis are shown below (**Fig. R2**). As predicted by the Reviewer, the C-ter directed anti-TAF1 antibody resulted in a substantially higher fraction of nuclear signal compared to the N-ter TAF1 antibody, further substantiating that fully synthesized TAF1 is more enriched in the nuclei. In accordance with the Reviewer's point, we amended our Discussion (Main text, page 18), but, we did not think that this Figure would add substantial information to our main message. If absolutely required, we could include the below **Fig. R2** in the revised manuscript.

Figure R2. TAF1 immunofluorescence experiment on HeLa cells. Two different TAF1 antibodies were employed as indicated. The nuclear and cytoplasmic fluorescence intensities were quantified in 3D and their ratio is plotted on the right.

Minor remarks

- The authors use immunofluorescence to score protein colocalization with mRNAs (Fig. 2-3). It is likely that the corresponding protein spots correspond to several co-localized polypeptides. In this respect, the transcripts that are not detected as co-localized with protein spots could also be translated, yet to an extent that does not allow detection of the corresponding protein (or partners). This should be indicated in the Results section.

We agree with the Reviewer's comment. We also noticed that not all co-localized protein spots show the same intensity. Although this might be due to the non-homogeneity of secondary antibody labelling, it might also capture different degrees of mRNA translation (i.e. number of engaged ribosomes). As required, we clarified this point in the Results section (Main text, page 8, end of the first paragraph).

- Could the authors comment on the fact that the TAF4/TAF1 mRNA interaction is partially insensitive to puromycin treatment (Fig. 3C)?

As required, we noted this point in the Results section (Main text, page 9).

- The authors indicate that the TAF1 protein encompasses several predicted ribosome pause motifs (Extended Data Fig. 6). Could they analyze available ribosome footprinting datasets to confirm the existence of such pause sites?

We thank the Reviewer for this excellent suggestion. Due to the inherent low Ribo-seq coverage on *TAF1* mRNA (a lowly expressed transcript), we decided to merge the available datasets to perform the analysis. As requested, we now updated our original Extended Data Fig. 6, now **Extended Data Figure 7** in the revised version, with a ribosome foot-printing meta-plot. Although we stress that due to the inherent heterogeneity of the merged datasets this analysis can only provide a general overview of the translation dynamics along *TAF1* mRNA, we can make the following observations: a) The ribosome occupancy is not homogenous along the transcript; specifically, a wide region roughly located between positions 1000-2000 (transcript numbering) shows very low and sparse signal, hinting at fast translation rates with no signs of ribosome pausing (see new **Extended Data Figure 7** and **Fig. R3** below). This region encompasses all three TAF6-binding motifs (T6BM i, ii and iii) and its fast translation might be critical to rapidly expose these interaction motifs for the co-translational recruitment of the respective TFIIID building-blocks. b) Downstream to this region, translation seems to significantly slow down (higher ribosome occupancy). This seems to occur once the synthesis of the heavily structured DUF3591 central domain has started. In this view, contrary to the simple short linear motifs represented by the T6BMs, the co-translational folding of DUF3591 domain with TAF7 might benefit from a slower translation rate. Although the three predicted pausing motifs originally included in the corresponding extended figure do coincide with local peaks of ribosome occupancy (**Fig. R3**), they can only partially explain the observed ribosome foot-printing pattern. Thus, we removed the predicted pausing motifs from the revised new **Extended Data Figure 7** (original Extended Data Fig. 6). Overall, we feel that new analysis in **Extended Data Figure 7** shows a general ribosome occupancy distribution that is more representative of TAF1 protein translation. Importantly, the merged ribosome occupancy distribution is fully compatible with our Co-TA model for TFIIID assembly. In the light of these new observations, we revised the corresponding section in the Discussion (Main text, page 19-20).

Figure R3. Positions of predicted pause sites and *TAF1* ribosome footprinting metaplot.

Ribosome occupancy meta-profile of human *TAF1* derived from merging all available Ribo-seq datasets present in RiboCrypt browser. Footprints signals coming from reading frames 2 and 3 are omitted for clarity. Below, the *TAF1* functional domains are aligned to the CDS. Protein numbering matches the transcript used for this analysis (ENST00000373790). Predicted pause motifs defined according to Ingolia et al., 2011 are indicated above as in the previous version of the figure. RFP, ribosome footprints.

- TAF8>TAF10 (lines 92, 133) should be TAF10>TAF8.
- Fig. S1C (line 164) should be Extended Data Fig. 1C.

We thank the Reviewer for pointing out these mistakes. We corrected the text accordingly.

Reviewer #2

Remarks to the Author:

The study is novel, comprehensive and impressive. The manuscript is well-written and the conclusions drawn are based on multiple lines of evidence. I anticipate it will be of great interest to scientists interested in transcriptional regulation and in co-translation in general. Multiple state-of-the-art complementary methodologies were employed. The quality of the data and presentation is high. Appropriate statistical testing was employed in the mass spectrometry analysis.

We thank the Reviewer's positive feedback on our "novel, comprehensive and impressive" manuscript. We highly appreciate her/his comments regarding the "great interest to scientists interested in transcriptional regulation and in co-translation" that our manuscript will raise.

Suggestions for future studies [...]

We thank the Reviewer for her/his thoughtful suggestions, many of which we already started to investigate in the light of future studies.

Major Comments:

1. It is important to demonstrate the specificity of TAF1 depletion (Figure 6), i.e., does knockdown (KD) of other TAFs result in similar observations?

We thank the Reviewer for raising the important point on the specificity of TAF1 KD effects. To address this, we performed KD experiments against other two TAFs: TAF4 (a core TFIID subunit) and TAF7 (the best characterized TAF1 direct partner). As already shown for TAF1, we subjected the cells to cytoplasmic/nuclear extracts separation. In our revised manuscript we present our novel results in new **Extended Data Fig. 6a**, together with the revised text in the Results section (Main text, page 14). We find that TAF4 KD had no apparent effects on other TAFs distribution between the two compartments. TAF7 KD led to a very modest cytoplasmic accumulation of TAF4 and TAF6. Neither TAF4 KD or TAF7 KD matched the extent of TAF1 KD on TFIID subunits redistribution demonstrating the specificity of the effect of TAF1 KD. Importantly, TAF1 distribution and levels were unaffected by the above mentioned TAF4 and TAF7 KDs. All these new results together further confirm the central role of the TAF1 subunit for TFIID complex assembly and nuclear import.

2. Although the authors mention that each GFP-tagged TAF was previously shown to incorporate into TFIID purified from nuclear extracts, N-terminal tagging of specific TAFs with GFP may be problematic, as some of their functional domains are N-terminally located, e.g., a protein kinase domain in the N-term. of TAF1, which was shown to phosphorylate RAP74 (PMID: 8625415), and histone fold domains

in the N-term. of TAF8 and TAF13. Could N-term tagging affect the function or perhaps, the stability of some of the TAF polypeptides, especially the small TFIID subunits?

We understand the concerns of the Reviewer's regarding potential alterations of protein function by N-terminal GFP-tagging. As already demonstrated in reference 26, all the GFP-tagged TAFs are able to successfully integrate into holo-TFIID according to GFP-IP coupled to MS, including the small TFIID subunits (see the corresponding Figure from ref. 26 in **Fig. R4**, below). As, we were also worried about the potential burden of the N-terminal GFP-tags on the function of the corresponding subunits, we limited the use of this system to the results described in Fig. 1d-e, where the N-terminal GFP tag was key to detect nascent polypeptides in parallel in all our RIP-based pull-downs. Note that in all other experiments we investigated the endogenous subunits.

Figure 6. Consistency in the relative abundance of nuclear DNA-free TFIID subunits. Relative abundance of identified TFIID subunits, normalized to TAF1. Modules are highlighted in dark grey (bait) and grey (partners); pink box-frames indicate average enrichment of each subunit. Data are derived from iBAQ values mean \pm s.d. of technical triplicates.

Fig. R4 GFP-fusion TAFs can integrate into holo-TFIID complex. Extract of Figure 6 from Ref 26.

3. TAF1 depletion leads to accumulation of TFIID subunits in the cytoplasm. It would be interesting to discuss these findings with regards to outcomes of TAF1 KO that were previously reported in the literature.

As requested, we have added one sentence to the revised Discussion section, pointing out that both in yeast and in metazoans *TAF1* gene is essential (Main text, page 19).

Minor comments:

1. p. 14 line 376 (Figure 6A) - TAF9 does not appear to be significantly enriched in the cytoplasmic extract following siRNA-mediated TAF1 KD. It is advisable to include a panel with quantitation of the protein levels based on multiple experiments, in addition to representative western blots.

The Reviewer is right in that TAF9 accumulation is not as evident as for other TAFs in the western blot. We rephrased the text by removing TAF9 from the list of affected subunits to avoid a potential overstatement (page 14). Regarding the quantification of western blots signals, we restrained ourselves from deriving quantitative conclusions. We feel that commenting on the quantitatively different degrees of accumulation among TFIID subunits may lead to overinterpretation of the results. In addition, our qualitative observations are backed up by MS results, obtained through a totally different detection method (see **Fig. 6b**).

2. The authors should indicate the difference between the inhibitory effects of cycloheximide (which “freezes” translating ribosomes on the mRNA) and puromycin (which releases nascent peptides from ribosomes) on protein synthesis.

We thank the Reviewer for this important suggestion. We included a better description on the effects of the two translation inhibitors in the Results section (Main text, page 5 first paragraph) and figure legends (see legend of **Fig. 1**).

3. p.9 line 260 – it would be better if the list of TAF4-interacting TAFs would be ordered based on NSAF values, rather than ascending numerical order.

As required, we changed the text accordingly (page 10).

4. For the benefit of readers who are unfamiliar with the pLDDT scores, the authors should relate to the pLDDT values (e.g. pLDDT > 90 are expected to be modelled to high accuracy, pLDDT between 70 and 90 are expected to be modelled well and pLDDT between 50 and 70 are low confidence and should be treated with caution).

Thank you for pointing this out. Now we better defined pLDDT and thus, we added a brief description of this scoring system (see legend of **Fig. 5b**).

5. p.14 line 368: I suggest rephrasing: “This would allow TFIID assembly on the N-terminal half of TAF1 before the protein is released from the ribosome.” to: This would potentially allow TFIID assembly on the N-terminal half of TAF1 before the protein is released from the ribosome.

We rephrased the text according to the Reviewer’s suggestion (page 13).

6. It would be interesting to discuss the identification of distinct TAF4B- and TAF9B-containing complexes (Figure 4).

We thank the Reviewer for drawing our attention to this issue. In the cytoplasmic anti-TAF4 IP we find TAF4 but not TAF4B peptides, suggesting that the isolated cytoplasmic TAF4-containing building block is either lobe A, or lobe B subcomplexes (as indicated in **Fig. 4g**), containing a single TAF4 copy. In contrast, in the TAF4 IP-ed nuclear sample we find both TAF4 and TAF4B peptides (**Extended Data Fig. 4a**, originally 3A), hinting at the isolation of the holo-TFIID, that contains two copies of TAF4 (i.e. TAF4 and/or TAF4B), incorporated in lobes A and B. We have now detailed these observations on page 10. Regarding TAF9 and TAF9B we cannot draw similar conclusions, as we do not have IP grade TAF9 antibodies.

7. p.13 lines 338-339 – reference to PMID: 8170939 should be added.

We thank the Reviewer for suggesting this important reference. We added it to the reference list.

8. Typos:

- p.11, 13, 39, Figure 5: pLDTT should be replaced by pLDDT

- p. 16 line 424 add: it - we find it remarkable...

- p.16 line 425 instead of: This point - This points

- p. 35 Figure legend: D Same as in C (not B)

We apologize for these mistakes. We changed the text to correct the above typos. We thank the Reviewer for pointing them out.

Reviewer #3

Remarks to the Author:

The model, as well as the novel role for TAF1, however, should be of broad relevance to the scientific community.

We are happy to learn that the Reviewer appreciated all the aspects of our work and that he also considered it of broad relevance.

Overall, the data are well presented and conclusions are largely supported by multiple independent lines of experimental evidence. My two major concerns with this manuscript are that 1) the figures would be easier to follow if they would contain a little more description and that 2) it is hard to evaluate the relative importance of measurements.

1) The authors make use of a broad array of methods, which makes the manuscript both elegant and convincing. But it also makes it likely that readers are not familiar with the one or another approach. To help readers staying engaged simplified descriptions of how the assays work would help. Specifically: A brief description that puromycin terminates translation while cycloheximide stalls it would make Fig.1 more instantly understandable for folks outside the field without the need to read the text or consult google. Fig.2C “white dots indicate an overlap of TAF1 protein and mRNA, either TAF1 or negative control CTNNB1”. [Site note: In Fig 3, I find the zoom out way more informative, potentially due to printing quality. Consider zooming in?]. 5B: I am not sure a reader should be expected to know what pLDTT is? Maybe add a sentence why you think this is important and what it does. If space is limiting, maybe the discussion could be shortened a little to accommodate 1-2 more sentences per figure.

We agree with the Reviewer that the brevity of our figure captions might limit the immediate understanding of our results. Thus, as required, where appropriate, we have expanded and changed the figure legends. In addition, we thank the Reviewer for pointing out some of our rather simplified descriptions. To further improve our manuscript, we addressed all the additional specific points raised by the Reviewer. We noted that several of them were in common with comments of Reviewer 2, being important for the “readability” of our work.

2) To improve evaluation of the relative importance of RIPed mRNAs it would be helpful to supplement qPCR with a less biased approach. One option could be to subject the IPed RNA to total RNA-seq (Ribo0). This approach could also reveal the absolute rank of TAF1 mRNA in each IP and if there may be another potentially non-TAF that may have been missed with qPCR. This or any other means to help better evaluate how impactful the measures are would help. I.e. how significant is 3% TAF1 in Fig1 F/G ?

This issue of “relative importance” is also relevant to Fig. 3. TBP, for example, shows a very low colocalization with TAF1 mRNA, despite binding N-terminal and thus potentially “earlier” co-translationally than TAF7. This could be as TBP also functions in other subcomplexes. Yet TAF7 is in 40% and TAF4 10% etc. What conclusions can we draw from this quantification? May it be more informative to plot what % of TAF1 mRNA has the IF protein (i.e. flip it?).

The Reviewer is raising an important point. The “relative importance” mentioned by the Reviewer would be very hard to tackle with our present tools, especially to quantitatively compare different subunits of the same complex. Several layers of potential experimental biases (different antibody affinities, different IP and amplification efficiencies, different target protein/RNA abundances) would make the attempt of quantitative comparison among subunits poorly reliable, even by employing a mRNA sequencing approach. We agree that the sequencing of the co-RIP-ed mRNAs may have certain interesting advantages, but it will not help us to overcome the differential GFP-tagged subunits expression level induced quantitative biases. We managed to remove the antibody bias from our systematic RIP approach by exploiting the common GFP tag. Yet, the exogenous nature of this system makes the quantitative comparison between different subunits expressed at different levels (i.e. different cell lines) hardly achievable. Being aware of this, we refrained from deriving any significant biological conclusion by overinterpreting the different quantifications obtained from our experiments (e.g. interpreting high vs modest RIP enrichment levels; see also our response to Reviewer 1 Major point 1 and Minor Remark 1).

Concerning the last point of the Reviewer concerning plotting “*what % of TAF1 mRNA has the IF protein*”, this is exactly what we did in our analyses (**Fig. 2-3**). All quantifications of the imaging experiments refer to the “fraction of mRNA spots co-localized with protein spots”.

Additional comments:

An exciting, somewhat left open question is whether TFIID co-translational assembly is ordered (controlled), random/chaotic or a mix of both. Some subcomplexes, i.e. TAF6-9,11-13,2-8 appear to be co-translationally reciprocal while most seem directional. The reciprocal nature of some may argue that certain interactions just occur early and by chance and have a low off rate. If co-translational assembly is evolutionary favored, one may expect to find the protein interaction domains enriched in N-terminal regions, rather than C-terminal ones. Furthermore, analogous to colinear expression of HOX gene clusters, if co-translational assembly is crucial, it could be expected that proteins with multiple interaction domains like TAF1 have them in the order of co-translational assembly. Last, I may be wrong here but it appears that there was more success tagging TAFs N-terminal than C-terminal. Can the authors comment how these observations align with their model? Does their domain analysis resonate with this speculation?

Based on recent bioinformatics analyses, during evolution proteins that assemble co-translationally have sustained large N-terminal interfaces in order to promote co-translational subunit recruitment (Ref 34). In agreement, out of the eight larger subunits of TFIID that participate in Co-TA as nascent polypeptides (TAF1, 2, 3, 4, 6, 7, 8, 9), seven have their interaction domains located in the N-terminal side of the given subunit (TAF1, 2, 3, 6, 7, 8, 9). This point is now mentioned in the Discussion section (Main text, page 17-18).

Given the authors expertise, I feel it would also be appropriate to ask the authors to discuss how far their finding aligns with (their and others) previously reported non-canonical TFIID complexes (i.e. in stem cells).

All our present data is in perfect agreement with the published structural TAF interactions and the outstanding cryo-EM TFIID structures. As a consequence, our assembly pathway, that acknowledges the existence of TFIID subcomplexes, is also in agreement with previously published descriptions of partial TFIID assemblies. This point has been reinforced in the Discussion section (page 17-18). Note however, that certain older studies based only on none-verified antibody detection methods may have come to conclusions that today do not fit in the generally accepted TFIID interaction schemes. Thus, it is difficult to comment on them.

Acknowledging that translation goes N to C terminal, given the importance the results have on the authors conclusions, I wonder if it was attempted to use the N terminal GFP-TAF1 or the antibody from 2A to confirm the findings in 4F. Would the authors now detect TBP? 11/13?

We have already published that in the GFP-TAF1 exogenous expression system an anti-GFP IP coimmunoprecipitates TBP from polysome extracts, in good agreement with our model showing that TBP is binding to the N-terminal TAND domain (see Ref 24).

Moreover, as suggested by the Reviewer, we tested the TAF1 N-ter antibody in immunoprecipitation using cytoplasmic and nuclear extracts from HeLa cells. As shown below, this antibody did not work for this application. The enrichment of endogenous TAF1 signal is barely detected in the IP, even in the nuclear fraction, with no signs of co-purification with other TFIID subunits. As stated in the text (Main text, page 11), we interpret our findings from **Fig. 4f** as suggesting that the levels of fully synthesized TAF1 in the cytoplasm are below of the detection limits of the mass-spectrometry method used, in contrast to the nuclear fraction (see also our response to Reviewer 1, Major Point 2).

Fig. R4 Immunoprecipitation test using TAF1 N-ter-directed (ab188427, lower panel) antibody on HeLa cells cytoplasmic and nuclear extracts.

Maybe consider drawing the grey ball in 4A as a multi protein complex, i.e. several balls?

We modified the figure as suggested by the Reviewer to remind the reader of the complex nature of the assemblies.

The list of primers and used antibodies with # is very useful. The detailed method section is laudable!

We thank the Reviewer for acknowledging our efforts.

Decision Letter, first revision:

Message: Our ref: NSMB-A46922A

5th Apr 2023

Dear Dr. Tora,

Thank you for submitting your revised manuscript "Hierarchical TAF1-dependent co-translational assembly of the basal transcription factor TFIID" (NSMB-A46922A). It has now been seen by the original referees and their comments are below. The reviewers find that the paper has improved in revision, and therefore we'll be happy in principle to publish it in Nature Structural & Molecular Biology, pending minor revisions to satisfy the referees' final requests and to comply with our editorial and formatting guidelines.

We are now performing detailed checks on your paper and will send you a checklist

detailing our editorial and formatting requirements in a couple of weeks. Please do not upload the final materials and make any revisions until you receive this additional information from us.

To facilitate our work at this stage, it is important that we have a copy of the main text as a word file. If you could please send along a word version of this file as soon as possible, we would greatly appreciate it; please make sure to copy the NSMB account (cc'ed above).

Sincerely,
Sara

Sara Osman, Ph.D.
Associate Editor
Nature Structural & Molecular Biology

Reviewer #1 (Remarks to the Author):

The revised manuscript by Bernardini et al. has now addressed all my previous comments, providing new experimental data or analyses, as well as additional discussion elements. Overall, the manuscript contains important new information on the assembly of multiprotein complexes, a set of results that will be of interest to a wide audience. I fully support the publication of this revised version in Nature Structural & Molecular Biology.

Reviewer #2 (Remarks to the Author):

The authors have appropriately addressed the comments. I recommend accepting the paper.

Very minor comments:

The authors did not correct 2 typos:

- replacing pLDTT by pLDDT within Figure 5b.

- correcting the legend of Extended Data Figure 1 - d Same as in (b) - should be: d Same as (c)

Reviewer #3 (Remarks to the Author):

This great manuscript has been further improved. Thus, although I stand with that RNA seq could have helped (even if you just compare within one RIP, it would be good to know what else is coming down that may not be looked for by qPCR), i think this point should not be in the way of getting this work published at NSMB.

Author Rebuttal, first revision:**Response to the reviewers' comments**

Please ensure you have addressed the remaining outstanding comments from Reviewer #2:

Very minor comments:

The authors did not correct 2 typos:

- replacing pLDTT by pLDDT within Figure 5b.
- correcting the legend of Extended Data Figure 1 - d Same as in (b) - should be: d Same as (c)

We addressed the remaining minor points of Reviewer #2. Specifically, we corrected the pLDDT labeling in Figure 5b, Extended Data Figure 5e and the corresponding figure legend. We thank the Reviewer for pointing out these typing errors.

Final Decision Letter:

: 31st May 2023

Dear Dr. Tora,

We are now happy to accept your revised paper "Hierarchical TAF1-dependent co-translational assembly of the basal transcription factor TFIID" for publication as a Article in Nature Structural & Molecular Biology.

Your paper will be published online soon after we receive proof corrections and will appear in print in the next available issue. You can find out your date of online publication by contacting the production team shortly after sending your proof corrections. Content is published online weekly on Mondays and Thursdays, and the embargo is set at 16:00 London time (GMT)/11:00 am US Eastern time (EST) on the day of publication. Now is the time to inform your Public Relations or Press Office about your paper, as they might be interested in promoting its publication. This will allow them time to prepare an accurate and satisfactory press release. Include your manuscript tracking number (NSMB-A46922B) and our journal name, which they will need when they contact our press office.

About one week before your paper is published online, we shall be distributing a press release to news organizations worldwide, which may very well include details of your work. We are happy for your institution or funding agency to prepare its own press release, but it must mention the embargo date and Nature Structural & Molecular Biology. If you or your Press Office have any enquiries in the meantime, please contact press@nature.com.

An online order form for reprints of your paper is available at <https://www.nature.com/reprints/author->

reprints.html"><https://www.nature.com/reprints/author-reprints.html>. Please let your coauthors and your institutions' public affairs office know that they are also welcome to order reprints by this method.

Please note that *Nature Structural & Molecular Biology* is a Transformative Journal (TJ). Authors may publish their research with us through the traditional subscription access route or make their paper immediately open access through payment of an article-processing charge (APC). Authors will not be required to make a final decision about access to their article until it has been accepted. [Find out more about Transformative Journals](https://www.springernature.com/gp/open-research/transformative-journals)

Authors may need to take specific actions to achieve [compliance with funder and institutional open access mandates](https://www.springernature.com/gp/open-research/funding/policy-compliance-faqs). If your research is supported by a funder that requires immediate open access (e.g. according to [Plan S principles](https://www.springernature.com/gp/open-research/plan-s-compliance)) then you should select the gold OA route, and we will direct you to the compliant route where possible. For authors selecting the subscription publication route, the journal's standard licensing terms will need to be accepted, including [self-archiving policies](https://www.springernature.com/gp/open-research/policies/journal-policies). Those licensing terms will supersede any other terms that the author or any third party may assert apply to any version of the manuscript.

Sincerely,
Sara

Sara Osman, Ph.D.
Associate Editor
Nature Structural & Molecular Biology
